# Tightening-up methane plume source rate estimation in EnMAP and PRISMA images

Elyes Ouerghi[1,2], Thibaud Ehret[1], Gabriele Facciolo[1], Enric Meinhardt[1], Rodolphe Marion[2], and Jean-Michel Morel[3]

[1]Université Paris-Saclay, CNRS, ENS Paris-Saclay, Centre Borelli, France
[2]CEA/DAM/DIF, F-91297 Arpajon, France
[3]City University of Hong Kong, China

**Correspondence:** Elyes Ouerghi (eouerghi@ens-paris-saclay.fr)

**Abstract.** Reducing methane emissions from human activities is essential to tackle climate change. To monitor these emissions, we rely on satellite observations, which enable regular, global-scale tracking. Methane emissions are typically quantified by their source rate – the mass of gas emitted per unit of time. Our goal here is to estimate the emission source rate of methane plumes detected by hyperspectral imagers such as PRISMA or EnMAP. For this task, we generated a large synthetic dataset using Large Eddy Simulations (LES) to train a deep learning model. This dataset was specifically designed to avoid network overfitting with careful plume temporal sampling and plume scaling. Our deep learning network, MetFluxNet, does not require any wind information or a plume mask. Moreover, it accurately predicts the source rate even in the presence of false positives. MetFluxNet performs well on our dataset with a mean absolute percentage error (MAPE) of $8.3\%$ over a wide range of source rates from $500 \text{ kg h}^{-1}$ to $25000 \text{ kg h}^{-1}$. Notably, it remains effective at lower source rates, where background noise is typically high. To validate its real-world applicability, we tested MetFluxNet on real plumes with known ground-truth fluxes. The predicted source rates fell systematically within the 95% confidence intervals, demonstrating its reliability for real-world plume estimation. Finally, in a comparison with recent state-of-the-art methods, MetFluxNet outperformed the deep learning-based S2MetNet and the physics-based Integrated Mass Enhancement (IME) method.

## 1 Introduction

The global warming potential of a methane ($CH_4$) molecule is 80 times larger than the global warming potential of carbon dioxide ($CO_2$) over a 20-year period. Thus, reducing methane emissions from human activities is an effective strategy to curb climate change. About a third of $CH_4$ emissions related to human activities come from coal, oil, and gas infrastructures (Jacob et al., 2016; Özgen Karacan et al., 2025). Hence, a substantial part of human $CH_4$ emissions could be controlled or reduced. Here, we focus on point source methane emissions. This designates plumes containing a large amount of $CH_4$ but coming from a small ground surface. To monitor methane emissions from anthropogenic activities, multiple satellites have been launched into Earth's orbit over the past decades, enabling global-scale monitoring.

Monitoring atmospheric methane concentrations with satellite imagery started in the early 2000s with the SCIAMACHY instrument (Frankenberg et al., 2005) onboard ENVISAT. The low spatial resolution of $30 \times 60 \text{ km}^2$ allowed a global scale

analysis, but not the detection of localized emissions. The use of high-resolution hyperspectral satellites to detect methane point source emissions began in 2016 with the work of Thompson et al. (2016) on Hyperion, followed by GHGsat (Jervis et al., 2021). Techniques for detecting methane plumes were also developed in AVIRIS airborne campaigns. These campaigns made it possible to continue to develop existing atmospheric inversion methods (Thorpe et al., 2013), as well as using new methods such as the matched filter (Thompson et al., 2015) and its variants (Funk et al., 2001; Theiler, 2021). The study of methane plumes has also been extended to multispectral instruments such as Sentinel-2 (Ehret et al., 2021) and WorldView-3 (Sánchez-García et al., 2022). Recently, a new generation of hyperspectral imagers, including PRISMA (Cogliati et al., 2021) and EnMAP (Guanter et al., 2015), have also proved their ability to monitor point source emissions (Guanter et al., 2021; Roger et al., 2024).

Here, we address the task of estimating the emission source rate for methane plumes detected by high-resolution hyperspectral sensors such as PRISMA and EnMAP. Several methods have been designed to estimate the emission source rate from a single plume observation, such as the cross-sectional flux (Varon et al., 2018; Jacob et al., 2022) or the angular width method (Jongaramrungruang et al., 2019). One of the most popular methods is integrated mass enhancement (IME) (Frankenberg et al., 2016; Varon et al., 2018), which is in particular used to estimate the methane source rate of plumes in PRISMA and EnMAP images (Guanter et al., 2021; Roger et al., 2024). However, these methods often have a high error rate and rely on external data, such as wind speed, which can introduce up to $50\%$ uncertainty (Varon et al., 2018).

In recent years, methods using deep learning and in particular convolutional neural networks (CNNs) have been used for source rate estimation (Jongaramrungruang, 2021). Convolutional neural networks capture the spatial features of the plume and the amount of gas at the same time. The spatial features of the plume are particularly relevant for this problem, as they are correlated with the wind speed (Jongaramrungruang et al., 2019). Wind speed is a crucial component for source rate estimation because it characterizes the diffusion speed of the plume. The most common CNN architectures for source rate estimation are the classic U-Net (Bruno et al., 2024) or ResNet (Radman et al., 2023). This allows using pre-trained networks with weights learned on other datasets which do not necessarily contain satellite images. The weights learned from datasets such as ImageNet have proven useful for satellite images (Radman et al., 2023). All of these networks take as input a methane concentration map retrieved from a hyperspectral or multispectral image. Different methods are used to obtain this concentration map depending on the type of sensor. Some of the most used retrieval techniques are the matched filter (Theiler and Wohlberg, 2013) for hyperspectral images (Guanter et al., 2021; Roger et al., 2024) and the multiband-multipass (Varon et al., 2021) for multispectral images (Radman et al., 2023). Training deep neural networks requires large datasets. However, real plume datasets with known source rates are extremely rare, limited to a few specific sensors, and typically very small. Hence, it is common practice to train and test networks on simulated plumes produced with Large Eddy Simulations (LES) (Varon et al., 2021).

Here, we aim at developing a deep learning technique to estimate the emission rate of point-source methane plumes detected by PRISMA and EnMAP. Firstly, we present a new dataset of simulated methane plumes produced with LES. These plumes are then inserted into real EnMAP images to obtain a dataset with real background noise. Next, we detail the procedure used to retrieve the methane concentration. Then, we present the different architectures we trained and tested on different training and test sets. Lastly, we present experiments comparing our method with the state-of-the-art IME and with other deep learning

methods. The experiments were performed not only on our simulated data, but also on a dataset simulated by Varon et al.
(2021), and finally on real plumes with ground truth obtained in controlled methane release experiments (Sherwin et al., 2023b, a). This comparison allows us to verify the generalizability of our model.

## 2 Materials

### 2.1 Hyperspectral data

The method presented here is designed for high-resolution hyperspectral satellites. The images we work with are Level 1 (L1) images from PRISMA (Cogliati et al., 2021) and EnMAP (Guanter et al., 2015). Both of these satellites provide hyperspectral images with a spatial resolution of 30 m. The methane absorption bands are located inside the $1500 - 2450$ nm range, in the short-wave infraRed (SWIR) range. This range is covered by the spectral channels of both PRISMA and EnMAP. In the SWIR, the spectral resolution of PRISMA varies between 9 nm and 15 nm and the spectral resolution of EnMAP is approximately 10 nm.

The deep learning models presented here were trained on simulated plumes. To train our networks, we inserted those plumes in true EnMAP L1 images to reproduce plumes with real background noise. The result of the LES procedure is an enhancement map providing the enhancement in methane concentration for each pixel. From this simulated enhancement map, we compute the plume transmittance at each wavelength by using a radiative transfer model and methane absorption cross-sections from the HITRAN data base (Gordon et al., 2017). The plume transmittance is then convolved with the instrument's spectral response function to obtain a spectrum for each pixel of the plume. The resulting spectrum is then multiplied with the radiance spectrum from the L1 data to obtain the simulated plume.

We used 48 background samples from different locations in North America, Middle East, and North of Africa. Those three areas are places where methane plumes are frequently detected with PRISMA and EnMAP (Guanter et al., 2021; Roger et al., 2024) and will therefore allow us to recreate real conditions. The different locations from which the background samples were taken are summarized in Table 1.

These areas allow us to have homogeneous and heterogeneous backgrounds in our dataset. We compare three of these background samples in Figure 1, where methane enhancement maps are computed using the matched filter technique (Guanter et al., 2021). The enhancement maps show mainly noise, but also retrieval artifacts associated with elements of the original scene, such as a road for the Uzbekistan image. By comparing the distributions in $\Delta$XCH4 between the three scenes we can observe three Gaussian distributions with a mean close to zero but with very different standard deviations. This is caused by the different types of surface between the scenes. In particular, a very heterogeneous scene can lead to a noisy enhancement map. Source rate estimation will be harder on those backgrounds as the signal-to-noise ratio between the plume and the background will be lower. This means that we will have fewer spatial features of the plume available in those cases. Including different background distributions in the dataset will help the network to be robust to different noise levels.

**Table 1.** List of the locations used to extract background samples for our synthetic dataset.

| Location | Latitude | Longitude |
|----------|----------|-----------|
| Algeria | 30.29 | 7.65 |
| Iraq | 30.48 | 47.38 |
| Kazakhstan | 45.15 | 52.77 |
| Saudi Arabia | 20.87 | 41.66 |
| Oklahoma (USA) | 35.42 | -99.03 |
| Kansas (USA) | 38.40 | -101.53 |
| Utah (USA) | 38.24 | -109.38 |
| Uzbekistan | 38.55 | 65.99 |

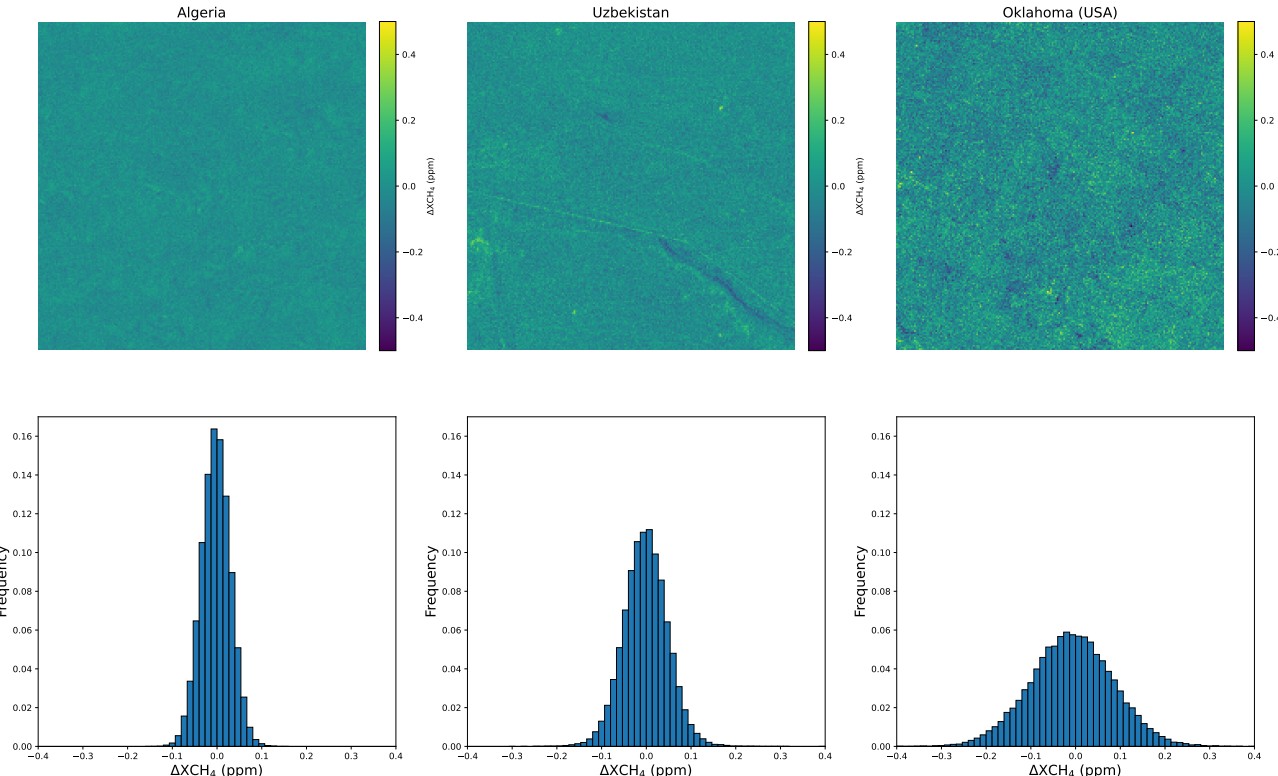

**Figure 1.** Comparison between different background samples and their corresponding distribution in terms of $\Delta$XCH4. The methane enhancement maps are estimated with the matched filter.

## 2.2 Large Eddy Simulations

Training a deep learning model requires a large amount of data. One of the main constraints in source rate estimation from satellite imagery is the lack of ground truth, which prevents us from using a dataset of real images. Therefore, we used our real plume images for testing purposes only. To train the model, we built a dataset of simulated plumes. To complete this dataset, we used the plume dataset generated by Varon et al. (2021) as a testing only dataset.

We created a dataset of simulated methane plumes with large eddy simulations (LES). The LES procedure allows one to simulate realistic plumes exposed to wind turbulence. We used the MicroHH model (van Heerwaarden et al., 2017) which has already been used for methane plume simulations (Ražnjević et al., 2022). We simulated at a spatial resolution of 30 m in a $6 \times 6$ km$^2$ domain. We used 61 different wind speeds between 0.5 m s$^{-1}$ and 6.5 m s$^{-1}$, and for each wind speed, we conducted 4 simulations with different temperature profiles, resulting in 244 different simulations. Each simulation lasted 3 hours, the first hour being used as spin-up. In the remaining two hours, we sampled one plume every 2 min. Our dataset thus contains 14640 methane plumes and we used 12444 of them for training and 2196 for validation. Splitting the dataset before data augmentation ensured that the network would not see a plume with exactly the same shape during training as during testing. The dataset was randomly split from the combined 14640 plumes and not from the 244 simulations. This ensures that for each wind speed, we have plumes simulated with different temperature profiles in the train set. During the simulation process, all plumes were generated with the same constant emission source rate.

As mentioned above, to verify that our model can generalize to a diversity of plumes, we also tested it on the simulations performed in Varon et al. (2021). The dataset of Varon et al. (2021), originally designed for Sentinel-2, was generated with WRF-LES (Skamarock et al., 2008). It contains 1200 methane plumes simulated at various wind speeds ranging between 1.5 m s$^{-1}$ and 5 m s$^{-1}$, and at a 25 m horizontal and 15 m vertical resolution over a $9 \times 9 \times 2.4$ km$^3$ domain. The simulations were obtained from five different wind speeds and sampled with a 30 s time gap. Before testing, the plumes were resampled at a resolution of 30 m with cubic splines interpolation. We will refer to this dataset as S2Test.

## 2.3 Source rate scaling

To study the performance of our model with a wide range of source rates, we performed data augmentation by randomly scaling all the plumes in the dataset. The plumes in the train set were generated from the result of the LES by randomly scaling each plume ten times to obtain source rates between 50 kg h$^{-1}$ and 33000 kg h$^{-1}$. It is very difficult to detect emissions at a source rate of 50 kg h$^{-1}$ with satellites such as PRISMA or EnMAP (Jacob et al., 2022; Cusworth et al., 2019). However, neural networks usually suffer from a threshold effect associated with the training range that prevents them to predict values outside said range. This leads to a bias in the predictions near the limits of the training range. Therefore, we propose to train the network on emission rates as low as possible. If we only considered source rates starting at 1000 kg h$^{-1}$, it would not be possible to know if a plume for which we estimate 1000 kg h$^{-1}$ is not actually at a lower source rate. Training from 50 kg h$^{-1}$ up ensures that the plumes that can actually be detected will not suffer from the threshold effect, the detection threshold for EnMAP being between 100 kg h$^{-1}$ and 500 kg h$^{-1}$ depending on the background (Cusworth et al., 2019).

The plumes in the test set were scaled between 100 kg h$^{-1}$ and 25000 kg h$^{-1}$. Due to the threshold effect described above, the range of source rates for the test set needs to be smaller than the range of source rates for the train set. Otherwise, the network would underestimate the source rate close to 33000 kg h$^{-1}$ and overestimate the source rate close to 50 kg h$^{-1}$. This would introduce a bias in the evaluation, which is avoided by evaluating on a, more realistic, narrower range.

## 2.4 Simulations temporal sampling

To generate our dataset, we used a time gap of 120 s between two plumes from the simulation, while in the dataset of Varon et al. (2021), the time gap is only 30 s. Datasets can even be found with shorter time gaps, such as the one used by Radman et al. (2023), which has a time gap of only 10 s. Increasing the time gap between simulated plumes in a dataset reduces their correlation and therefore increases the diversity of the dataset. If we consider plumes taken with a small time gap (less than 30 s), we can observe the same turbulence patterns; thus, they can hardly be considered as different samples in the dataset. We can observe this redundancy in Figure 2, where we show the same plume at different time steps and for different wind speeds. We can easily notice that after 10 s and for any wind speed, the plume is almost identical to the initial image, whether it is in terms of shape or concentration. After 30 s, the shape is still quite similar, but there are some changes in the distribution of the concentration. This observation is mostly true around the source of the plume. In 30 s, the new concentration distribution has not yet spread to the tail of the plume. After 60 s, the changes in the distribution of the concentration have increased and we start to see some noticeable changes at the beginning of the plume tail. This is visible for the plumes at 1 m s$^{-1}$ and 3 m s$^{-1}$. After 120 s, most of the plumes are globally different from their original image. However, we still see residuals from the turbulence that were occurring in the initial image. For example, for the plume at 2 m s$^{-1}$, even if the concentration distribution is different, the overall shape of the plume after 120 s looks similar to the one in the initial image.

Thus, using small time gaps leads to a low plume variety in the dataset. This can lead to severe network overfit. In addition, the train set and test set will be strongly correlated, and thus overfitting will be more difficult to notice. To show the overfitting effect caused by small time gaps, we generated a second dataset following the methodology of Radman et al. (2023): we performed one simulation per wind speed and used a time gap of 10 s. We will refer to the dataset with a large time gap as MicroL (for MicroHH-Large) and to the dataset with a short time gap as MicroS (for MicroHH-Short). The dataset of Varon et al. (2021) referred to as S2Test will be used only for testing. The parameters for each dataset are summarized in Table 2. Note that MicroL does not result from subsampling MicroS. Otherwise, we would not be able to show the overfitting when working with MicroS, as performing well on MicroS would also lead to performing well on MicroL. Therefore, there is no plume in common between MicroL and MicroS and the plumes come from different simulations. The MicroL dataset is publicly available for download at https://doi.org/10.5281/zenodo.15618044.

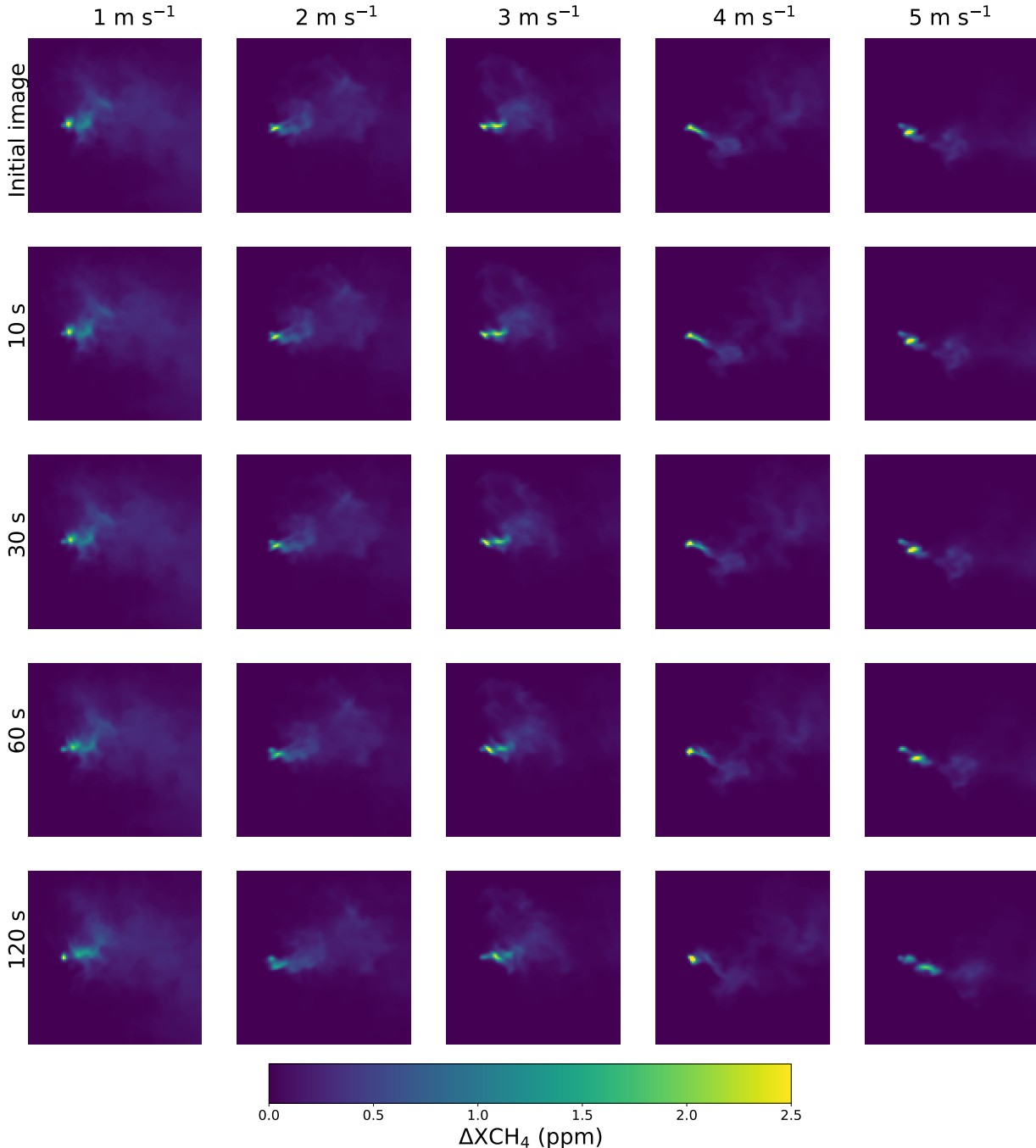

**Figure 2.** For different wind speeds a methane plume is displayed at different time steps. The plumes come from the proposed MicroS dataset. The images correspond to the result of the LES, before it is included in a real EnMAP image.

**Table 2.** Different test sets of simulated data. The number of train samples and test samples are the numbers before data augmentation. There is no train sample for S2Test as we use it only for testing.

| Datasets | Wind range | Temporal sampling | Number of different simulations | Number of train samples | Number of test samples |
|---|---|---|---|---|---|
| MicroL | 0.5-6.5 m s$^{-1}$ | 120 s | 244 | 12444 | 2196 |
| MicroS | 0.5-6.5 m s$^{-1}$ | 10 s | 61 | 37332 | 6588 |
| S2Test | 1-6 m s$^{-1}$ | 30 s | 30 | 0 | 1200 |

## 3 Methods

### 3.1 Methane concentration retrieval

In hyperspectral imaging, any object in a scene can be assigned a spectral signature. In the case of methane, this spectral signature is the absorption spectrum of the gas. To determine whether an observation contains an excess of methane, it is therefore natural to look for a deviation in the observation spectrum in the direction of the methane spectral signature. The amplitude of the observed deviation then provides a measure of the gas concentration. This idea sums up how the matched filter retrieval works for methane concentration. It is used on many hyperspectral instruments such as AVIRIS (Foote et al., 2020) and PRISMA (Guanter et al., 2021).

A standard hypothesis for hyperspectral images is that background pixels follow a Gaussian multivariate distribution (Theiler and Wohlberg, 2013). With this assumption, the maximum likelihood estimator of the methane mixing ratio is given by the matched filter (Huang et al., 2020).

Let us denote by $K_{CH4}$ the diagonal matrix whose diagonal components are the methane absorption coefficient values and let $\mu$ and $\Sigma$ be respectively the mean vector and the background covariance matrix. We define the target vector by

$$\mathbf{t} = -K_{CH4} \cdot \mu. \tag{1}$$

With these notations, the excess methane concentration $\alpha$ corresponding to an observed pixel $x$ is given by the matched filter formula:

$$\alpha(x) = \frac{\mathbf{t}^T \Sigma^{-1}(x - \mu)}{\mathbf{t}^T \Sigma^{-1} \mathbf{t}}. \tag{2}$$

Parameters $\mu$ and $\Sigma$ are computed with their empirical unbiased estimates. They are calculated across-track, which means that we compute a different set of parameters for each detector element in the sensor. This applies for both PRISMA and EnMAP images.

The matched filter is the optimal detector for an additive target in a Gaussian background. This assumption on the background is not necessarily true in methane plume detection. Several variations of the matched filter are designed to improve the quantification provided. Here, we use the MAG1C method proposed by Foote et al. (2020) for retrieving methane concentration.

The MAG1C method introduces two improvements to the matched filter formulation. The first is a spatial $L_1$ regularization to take into account the fact that most observations are not part of a plume. The second is the estimation of a different albedo coefficient for each pixel. The latter is defined by

$$r(x) = \frac{x^T \mu}{\mu^T \mu}.$$ (3)

This albedo coefficient is used to scale the target spectrum. Thus, the target spectrum used in the matched filter for pixel $x$
becomes $r(x)\mathbf{t}$ instead of $\mathbf{t}$.

The MAG1C method can be applied to images from both PRISMA and EnMAP. The main difference when applying it to those instruments is the target spectrum. To calculate the target spectrum, we need to compute a unit methane absorption spectrum through radiative transfer simulation (Guanter et al., 2021; Roger et al., 2024). This requires to compute convolutions with the spectral response function of the instrument, which is not the same for PRISMA and EnMAP. However, we obtain
similar enhancement maps when applying MAG1C to images from both satellites.

### 3.2 Integrated Mass Enhancement

The most classical method for estimating point source methane emissions is Integrated Mass Enhancement (IME) (Frankenberg et al., 2016). This method is already widely used for EnMAP and PRISMA images (Roger et al., 2024; Guanter et al., 2021). The source rate $Q$ is calculated as

$$Q = \frac{U_{eff} \cdot \text{IME} \cdot 3600}{L},$$ (4)

where the IME is the total mass of excess methane (in kg) contained in the plume, $L$ is the plume length (in m) and $U_{eff}$ (in m s$^{-1}$) is the effective wind speed. The factor 3600 results from the conversion from kg s$^{-1}$ to kg h$^{-1}$. The effective wind speed $U_{eff}$ is usually estimated from the wind speed at 10 m altitude $U_{10}$. The relationship between $U_{eff}$ and $U_{10}$ is obtained by fitting a regression model on simulated data with Large Eddy Simulations (Guanter et al., 2021; Varon et al., 2018). Several
expressions exist for $U_{eff}$ with linear or logarithmic models (Guanter et al., 2021; Varon et al., 2018). A model suited for source rate estimation with PRISMA or EnMAP is (Guanter et al., 2021; Roger et al., 2024)

$$U_{eff} = 0.34 \cdot U_{10} + 0.44.$$ (5)

In Equation 4, the IME is obtained from the estimated CH$_4$ concentration in the plume. The length of the plume is usually calculated by taking the square root of the plume area (Varon et al., 2018). This allows one to deal with the fact that the length
of the plume is not always properly defined. Indeed, because of turbulence and wind variations, the plume does not necessarily follow a straight path. However, this implies using a plume mask to compute $L$. Therefore, the quality of the estimation of $Q$ will depend on the quality of the mask. A good mask (one that provides a good estimate of $Q$) is difficult to obtain. The varying plume shapes and the amount of noise in the images make it difficult to distinguish the contours of the plumes. This means that two different human operators can label the same plume in very different ways. This can affect not only the quality of the
estimation of $Q$, but also the reproducibility of the method.

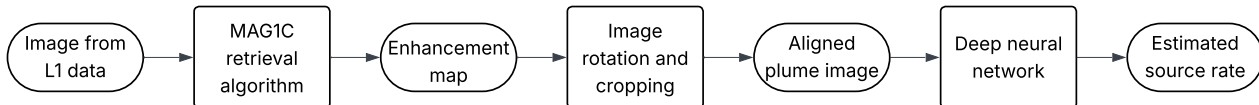

**Figure 3.** Diagram summarizing the different steps of our source rate estimation method with plume alignment.

To obtain $U_{10}$, a standard practice is to calculate it with an external measurement coming from a data set of wind data at a global scale such as GEOS-FP (Molod et al., 2012) or the ECMWF-ERA5 dataset (Hersbach et al., 2020). However, these datasets provide wind data with a low spatial resolution (around $25 \times 25$ km$^2$ for GEOS-FP and $30 \times 30$ km$^2$ for ERA5) and a low temporal sampling (hourly data). Hence, these datasets are not ideal as wind data sources to characterize CH$_4$ emissions: the temporal gap between the emission and the wind data point can be up to 30 minutes and most plumes studied with PRISMA or EnMAP will not exceed 5 or 6 km.

To compare our method with the IME, we considered two cases. In the first case, we estimate the source rate by using the effective wind given by Equation 5 obtained by Guanter et al. (2021) with LES simulations. In the second case, we fit our own effective wind model using the MicroL dataset. We obtain the following equation for $U_{eff}$

$$U_{eff} = 0.17 \cdot U_{10} + 0.49. \tag{6}$$

To fit our effective wind model, we use a different masking procedure than Guanter et al. (2021). We first apply a threshold to the plume image, corresponding to the quantile at $85\%$. Then, we apply a $4 \times 4$ mean filtering and a thresholding to the resulting image. To remove remaining potential false positives, we finally apply a $3 \times 3$ mean filtering followed by another thresholding. Once the set of plume masks is calculated, the wind model is obtained by performing a linear regression. Using different masking procedures explains why our model is different from the one used by Guanter et al. (2021). We will refer to our version of the IME as IME-MicroL.

### 3.3 Deep learning

To estimate the emission source rate from the methane concentration retrieval image, we used a deep neural network. The use of a neural network enables the estimation of the source rate without relying on an external data source for the wind speed. It also removes the variability associated with the manual labeling of the plume that is needed when using methods such as Integrated Mass Enhancement (Frankenberg et al., 2016). The different steps of the source rate estimation pipeline are summarized by the diagram presented in Figure 3.

#### 3.3.1 Models and training

We compared two architectures. Firstly, the EfficientNetV2-B0 (Tan and Le, 2021) model. This is the lightest version of the EfficientNet models in terms of number of parameters with 7.2 million parameters. Those models have already proven

their efficiency for source rate estimation (Radman et al., 2023). The use of the lightest version allows for a fast training, even on CPU. The second architecture tested is the ConvNeXt-Tiny (Liu et al., 2022) model. This is the lightest version of the ConvNext (Liu et al., 2022) models but it has four times more parameters than EfficientNetV2-B0 with 28.6 million parameters. For both models, we changed the last layer for a fully connected layer with one unit to perform the source rate estimation. Those models expect an input with three channels. To satisfy this condition, we converted the methane enhancement map into a three-channel image by copying it in each new channel. For the training, we fine-tuned the models weights that had been trained on ImageNet. We compared the Mean Square Error (MSE) loss and the Mean Absolute Percentage Error (MAPE) loss. During training, $15\%$ of the train set was used for validation. The validation set is a subset of the train set which is used to check that the network improves its predictions and does not overfit during the training. The network was not trained on the samples belonging to the validation set.

### 3.3.2   Dataset pre-processing

We used the two architectures described above to train several networks. These networks will allow us to compare different pre-processing for our plumes images such as rotations and shifts of the plumes.

The most common pre-processing consists in augmenting the dataset with random rotations of the plumes and random shifts of the source (from 0 to 3 pixels) in any direction, as in Radman et al. (2023). This creates datasets as diverse as possible and helps reproduce real plume images.

This pre-processing artificially increases the difficulty of the task. Applying random rotations to the plumes means that pixels forming the plume can be found everywhere in the image and the network will have to find them to extract meaningful information. Otherwise, the network might just use noisy pixels to compute its prediction, particularly in the case of plumes with a low source rate. This is likely to affect the quality of the source rate estimate.

However, in the context of plume quantification, we already know that our image contains a methane plume, and we know its position. By leveraging this knowledge, we aim to simplify the neural network's task. Since the plumes locations are known, our pre-processing involves aligning all the plumes with the x-axis, so that they propagates from left to right in the image. We then crop the rotated images to ensure that the source of the plume is located at the same fixed position for all the images.

In real-case scenarios, precisely locating the sources and perfectly rotating the plumes can be challenging. In order to process images containing a plume, we first apply a rotation corresponding to the opposite of the apparent angle of the plume, which does not necessarily match with the wind angle. Then, to find the source of the plume, we look for the apparent vertex of the conical shape that describes the shape of the plume. This process is done manually for real plumes and does not require knowledge of the actual plume source or wind angle.

When rotating and cropping the plume image, there are uncertainties associated with both operations. Hence, the pre-processing will not always be perfect. To account for this uncertainty, in the training set, we add random shifts to the plume source (from 0 to 3 pixels) in any direction. This ensures that the network will be robust to an inaccurate cropping of the image.

The other source of uncertainty arises from the alignment of the plumes with the x-axis, which can be imprecise. Consequently, the training set must include examples of plumes that have a non-zero angle relative to the x-axis. During the LES

procedure, eddies naturally cause direction changes, meaning that plumes do not always spread in the direction of the wind and may have a non-zero angle with the x-axis. This phenomenon is visible in Figure 2, where plumes in the last two columns are visibly misaligned with the x-axis. The training set includes plumes with angles ranging from $-70°$ and $-70°$ relative to the x-axis, ensuring that the network remains robust against inaccurate image rotations. Therefore, we do not artificially rotate simulated plumes to achieve this augmentation.

Thus, for the dataset with aligned plumes, we have uncertainties on both source location and plume alignment. The size of each image is $100 \times 100$ pixels, covering an area of $3 \times 3$ km$^2$. Because all of the plumes are now aligned, the source is placed on the left of the image.

We will compare this dataset with a dataset containing random rotations of plumes with angles between 0 and 360 degrees. For this dataset, the rotated plume is located at the center of the image. Like for the dataset with alignment, we apply random shifts (after the rotation) of the source. The size of each image is $130 \times 130$, covering an area of $3.9 \times 3.9$ km$^2$. This area is large enough to contain most of the plumes that are usually detected with PRISMA and EnMAP. If the plume is larger than the cropped image, the part outside of the frame corresponds to the end of the plume tail. For that part of the plume, the magnitude of the retrieved values is usually at the background level, so very little to no exploitable information can be obtained from this area.

## 4  Experiments and results

To evaluate the results, we use two standard metrics: the Root Mean Square Error (RMSE) and the Mean Absolute Percentage Error (MAPE). We are going to compare the method presented here with different source rate estimation techniques but also with the different datasets MicroL, MicroS, and S2Test. The test sets in these datasets contain plumes with source rates starting at 100 kg h$^{-1}$. However, in real-life conditions, it is highly unlikely that plumes with such low source rates will be detected, as they are below the detection threshold of PRISMA and EnMAP (Jacob et al., 2022; Cusworth et al., 2019). To calculate MAPE and RMSE, we will therefore only use plumes with source rates above 500 kg h$^{-1}$. Plumes with source rates below this threshold will still be used for visualization purposes to observe the networks' behavior at very low source rates.

### 4.1  Architecture and plume orientation

We start by studying the influence of the network architecture and of the plume orientation. To do so, we compare different networks for which we select an architecture between EfficientNetV2-B0 and ConvNeXt-Tiny and a plume orientation between random rotations and alignement with the x-axis as described in the previous section. This leads to four networks: EffNet+rotation, EffNet+align, CNext+rotation, CNext+align. These four networks are trained with MSE loss, on the dataset MicroL.

In Table 3, we compare the results of EffNet+rotation, EffNet+align, CNext+rotation, CNext+align in terms of RMSE and MAPE. Overall, the methods with plume alignment outperform the other networks both in RMSE and in MAPE. We can also observe that, for a fixed pre-processing, the networks based on ConvNeXt seem to perform better than those based on

**Table 3.** Result comparison of four networks tested on MicroL. The metrics are computed with the plumes in the test set belonging in the 500 kg h$^{-1}$ - 25000 kg h$^{-1}$ range.

| Name | RMSE | MAPE |
|------|------|------|
| EffNet+rotation | 1736 | 12.8 |
| EffNet+align | **1421** | 10.2 |
| CNext+rotation | 1551 | 10.3 |
| CNext+align | 1437 | **9.5** |

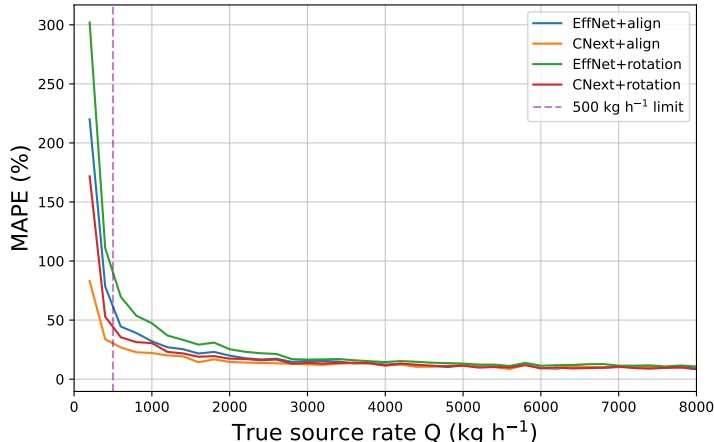

**Figure 4.** Evolution of the MAPE with respect to the source rate for different architectures and plume orientations. The networks are trained and tested on the MicroL dataset.

EfficientNet. Whereas it is clear that CNext+rotation outperforms EffNet+rotation, CNext+align outperforms EffNet+align only in MAPE. The networks with plume alignment have a very close RMSE with a gap of only 16 kg h$^{-1}$, which is not statistically significant. However, a gap of 0.7 in MAPE shows a real difference in performance. Indeed, the low source rates have very little impact on the RMSE but can have a high impact on the MAPE. A lower value in MAPE but not in RMSE therefore means that the estimation is improved for low source rates.

We can observe the evolution of the MAPE with respect to the source rate in Figure 4. We see that the four tested networks have very similar performance levels for high source rates, from 4000 kg h$^{-1}$ upwards. Above 4000 kg h$^{-1}$, the MAPE hardly decreases at all, remaining around 10% for all networks. Indeed, at high source rates, the methane concentration looks like the ground truth image (the output of the LES), meaning that the noise is negligible with respect to the plume concentration. Therefore, for the network, there is no difference (in terms of additional information) between an image with a plume at 10000 kg h$^{-1}$ or at 20000 kg h$^{-1}$. Thus, the networks differ only in their performance at medium- and low-source rates, with

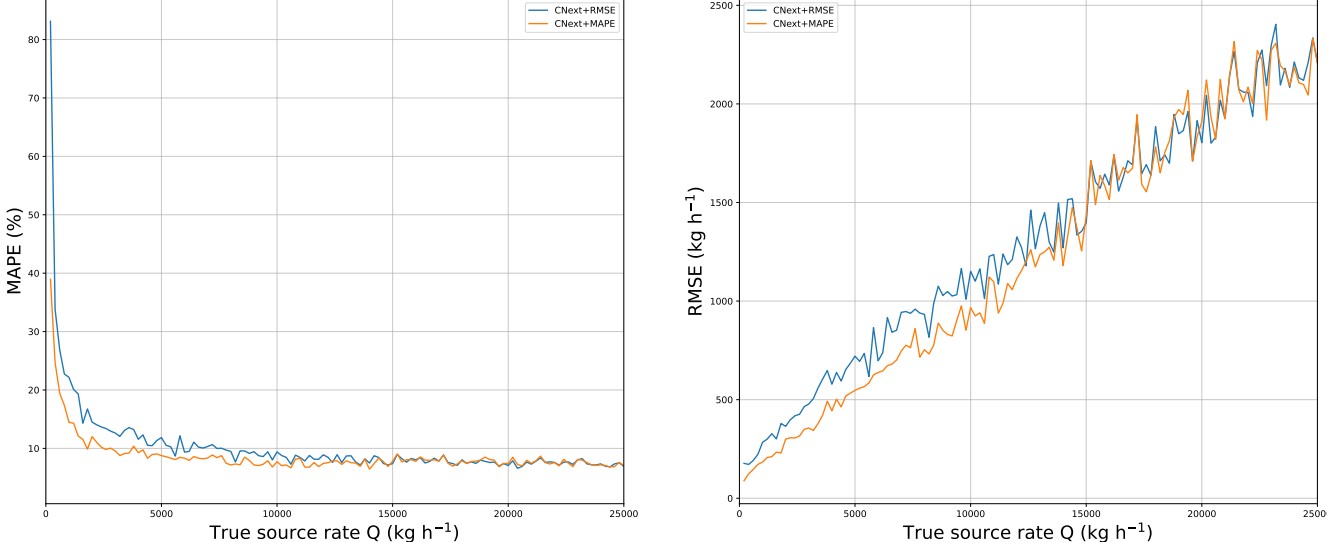

**Figure 5.** Evolution of the MAPE and the RMSE with respect to the source rate for networks trained with the MAPE loss and the RMSE loss. The networks are trained and tested on the MicroL dataset with aligned plumes.

the gaps between them narrowing as the source rate increases. In particular, we can see that CNext+align is outperforming EffNet+align for low source rates. Between 100 kg h$^{-1}$ and 200 kg h$^{-1}$, the MAPE of CNext+align is at least half the MAPE of any other network. However, even if we dismiss the case of source rates below 500 kg h$^{-1}$, CNext+align still outperforms the other methods. Since MAPE is a better representation of the networks performance over the entire range of source rates, from now on, we will focus only on the ConvNextTiny architecture.

## 4.2 Loss

As we saw in Figure 4, the differences in performance between the methods lie in the 0-4000 kg h$^{-1}$ range. However, when training with MSE loss, this is the range that has the least weight in the loss. To improve performance in the 0-4000 kg h$^{-1}$ range, we train the network directly with MAPE loss, which gives more weight to low source rates than MSE loss. This is also possible because performance in the 4000-25000 kg h$^{-1}$ range is stable for all the networks used, so we can expect the same result when changing the loss. In Figure 5, we observe the influence of the loss. We compared a network CNext+RMSE trained with the RMSE loss on aligned plumes with CNext+MAPE, the same network but trained with the MAPE loss.

As expected, CNext+MAPE outperforms CNext+RMSE in terms of MAPE. We can see a significant improvement in the 0-10000 kg h$^{-1}$ range. Note that changing the loss affects not only plumes with small source rates but also those with higher rates. Moreover, it also outperforms CNext+RMSE in terms of RMSE with a lower RMSE in the 0-10000 kg h$^{-1}$ range. Beyond 10000 kg h$^{-1}$, the two networks have similar performance.

The network CNext+MAPE trained with aligned plume on MicroL is the best version of our different networks and we name
it MetFluxNet.

### 4.3 Uncertainty estimation

The simplest way to estimate the uncertainty on the source rate estimate provided by the neural network is to compute the empirical standard deviation of the estimation. To compute it for a given prediction, we consider a sample of the true source rate distribution corresponding to this prediction and we compute the standard deviation of this distribution with its usual non biased empirical estimate. The sample of the true source rate distribution is obtained from the MicroL test set. Under the assumption that the source rate distribution corresponding to a prediction made by the network locally follows a Gaussian distribution, we can then obtain a confidence interval on the prediction.

Under the same assumption, another way to obtain a confidence interval is to train the network for a probabilistic regression (Nix and Weigend, 1994). For a given plume $P$, let us denote by $Q$ its emission source rate. Then, the prediction made by the network for $P$ follows a Gaussian distribution $\mathcal{N}(\hat{Q}, \sigma)$, where $\hat{Q}$ is an estimator of $Q$. When using a probabilistic regression, we want to estimate both $\hat{Q}$, which will be the predicted source rate, and $\sigma$ which will be the standard deviation of the estimation. This standard deviation yields confidence intervals.

Predicting the standard deviation requires a small change in the network architecture. The previous networks used a fully connected layer with one unit as the last layer to perform the source rate estimation. To output both the predicted source rate and the standard deviation, we add in parallel of this last layer a fully connected layer with one unit set to the power of two. Squaring the layer ensures that the output is positive. Therefore, we consider that the output of this second layer will be the variance of the distribution, that is, $\sigma^2$.

To ensure that $\sigma$ is an estimate of the standard deviation, we use the negative log likelihood (NLL) as loss. Indeed, if $(\hat{Q}, \sigma)$ minimizes the NLL, then $(\hat{Q}, \sigma)$ is the maximum likelihood estimator for the parameters of the output distribution of the network. The NLL is defined by

$$\text{NLL}(Q, \hat{Q}, \sigma) = \frac{1}{2} \left( \log 2\pi\sigma^2 + \frac{\left\| \hat{Q} - Q \right\|^2}{\sigma^2} \right). \tag{7}$$

We obtain a similar performance for the predicted source rate when comparing CNext+MAPE with CNext+NLL. The CNext+MAPE has a RMSE of 1388 kg h$^{-1}$ and a MAPE of 8.3% whereas CNext+NLL has a RMSE of 1369 kg h$^{-1}$ and a MAPE of 8.3%.

In Figure 6, we compare the empirical standard deviation computed with the CNext+MAPE output, denoted by $\sigma_{emp}$, with the network estimated standard deviation computed with CNext+NLL, denoted by $\sigma_{NLL}$. Note that the plot depends on the predicted source rate and not on the true source rate because we look at the distribution of the network output. We can notice that $\sigma_{NLL}$ and $\sigma_{emp}$ have the same behavior. The proximity between the values of $\sigma_{emp}$ and $\sigma_{NLL}$ is due to the fact that the empirical standard deviation is the maximum likelihood estimator of the standard deviation for a Gaussian distribution. Since

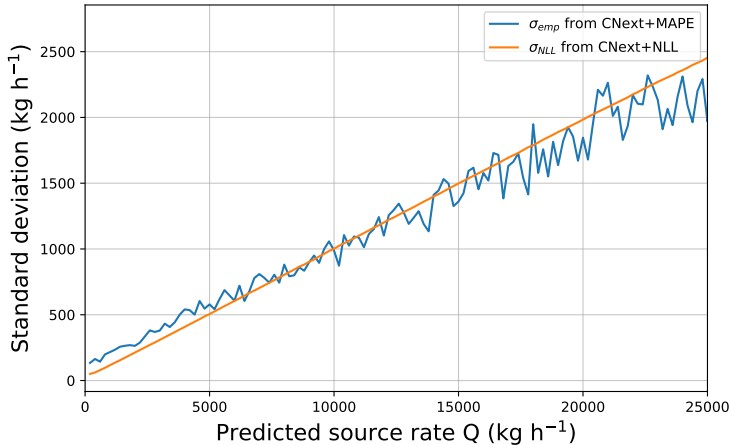

**Figure 6.** Evolution of the standard deviation computed empirically or with a probabilistic regression with respect to the predicted source rate. The networks are trained and tested on the MicroL dataset with aligned plumes.

$\sigma_{NLL}$ is an approximation of the maximum likelihood estimator of the standard deviation, we deduce that $\sigma_{NLL}$ should be close to $\sigma_{emp}$. From now on, we name ProbMetFluxNet the CNext+NLL network.

The standard deviations computed for our networks in Figure 6 are way lower than those provided by the IME for the same source rates (Guanter et al., 2021). This is mainly due to the fact that our neural networks do not rely on external data. With IME, a large part of the uncertainties comes from the wind speed. It is common to assume that there is around $50\%$ of error on the wind speed (Guanter et al., 2021). Hence, removing the wind speed from the estimation pipeline allows to significantly reduce the standard deviation on the estimation and therefore the size of the associated confidence interval.

### 4.4 Influence of the background

All the presented networks estimate the emission source rate using all the information in the image, including background data. However, the background can contain false positives, typically pixels not belonging to the plume that could be considered as plume pixels because of their high retrieved concentration. When estimating the source rate, we first want to remove those false positives before giving the image to the network. Removing a part of the background pixels will change the overall distribution of the background. In particular, the resulting distribution will be different from the ones the network has seen in training. This might lead to errors in the source rate estimation.

In Figure 7, we observe two images of the same simulated plume. The source rate corresponding to this plume is $2192\,\mathrm{kg\,h^{-1}}$. In the left image, we have the original methane retrieval image. In the right image, we removed the top and bottom edges, which contained only background pixels. Even if these images do not include false positives, this example shows how a change in the background far from the plume can impact the source estimation.

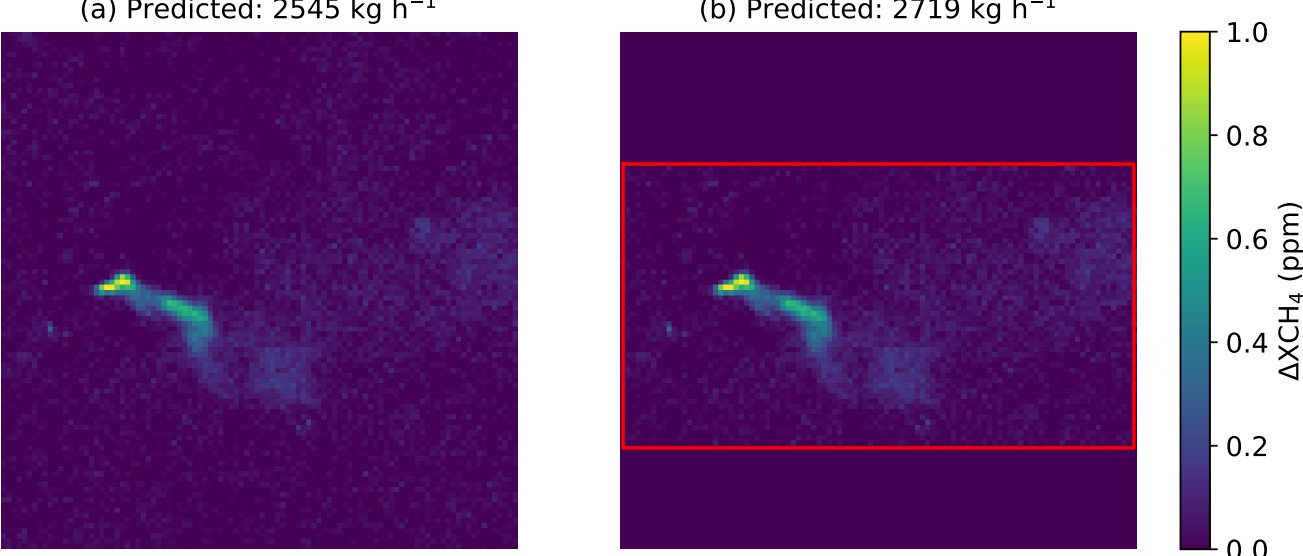

**Figure 7.** Two images of the same simulated plume. The left one is the result of the methane concentration retrieval. On the right one, we removed background pixels in the top and the bottom of the image. The true source rate corresponding to those plumes is 2192 kg h$^{-1}$. The scale is in particles per million (ppm).

We can see that although the two plumes are identical, we have two significantly different source rate predictions with an increase of almost $10\%$ in the predicted source rate when removing a part of the background pixels. Moreover, this increase in the predicted source rate widens the gap between the prediction and the ground truth.

To reduce the impact of the background distribution, we trained a version of our network while removing different parts of the background. This aims at reproducing the background distribution we would obtain when removing false positives in real plume images. To create those sparse images to train the network, we draw random bounding boxes that include the entire plume, and we remove the pixels outside of it. This avoids mistakenly removing plume pixels. The random bounding boxes are used during training to be robust to different bounding box sizes and therefore different levels of sparsity in the background distribution. The idea behind the use of bounding boxes is that it is easier to draw a bounding box than a fine mask of the plume. This comes with an important drawback: if a false positive is close to the plume, it might not be possible to remove it with a rectangular bounding box. However, this limit also exists for fine plume masks when the plume intersects a false positive.

In Figure 7, the right-hand image corresponds to the bounding box applied to the left-hand plume. We name MicroL-sparse the MicroL dataset with the partial background removal. In the same way, we name MetFluxNet-sparse the version of MetFluxNet trained on MicroL-sparse.

In Table 4, we compare MetFluxNet and MetFluxNet-sparse on MicroL and MicroL-sparse. As could be expected, MetFluxNet has the best performance on MicroL and MetFluxNet-sparse obtains the best performance on MicroL-sparse. In particular, the performance of MetFluxNet on MicroL is similar to the performance of MetFluxNet-sparse on MicroL-sparse.

**Table 4.** Result comparison for MetFluxNet and MetFluxNet-sparse. The networks are trained respectively on MicroL and MicroL-sparse. They are both trained with the MAPE loss.

| Network | RMSE (MicroL) | RMSE (MicroL-sparse) | MAPE (MicroL) | MAPE (MicroL-sparse) |
|---|---|---|---|---|
| MetFluxNet | **1388** | 1728 | **8.3** | 10.3 |
| MetFluxNet-sparse | 1445 | **1374** | 8.9 | **8.3** |

**Table 5.** Result of MetFluxNet on different backgrounds. The RMSE values are in kg h$^{-1}$.

| Area | RMSE | MAPE |
|---|---|---|
| North America | 1411 | 8.5 |
| Middle East | 1372 | 8.3 |
| North Africa | 1382 | 8.1 |

Hence, removing background pixels when there is no false positive to remove does not improve the results, but it does not decrease them either (the gap between a RMSE of 1388 kg h$^{-1}$ and 1374 kg h$^{-1}$ is not statistically significant). However, the results of MetFluxNet-sparse are much better on MicroL than the results of MetFluxNet on MicroL-sparse. This is because MetFluxNet-sparse is trained on images with various degrees of sparsity, therefore it generalizes better when there is no added sparsity in the images. However, MetFluxNet has the advantage of being able to be used without any manual intervention on the background.

Since MetFluxNet-sparse requires additional pre-processing to create bounding boxes and because those bounding boxes do not necessarily allow us to remove all the false positives, our main focus in the following sections will be on the standard MetFluxNet method.

Another way to look at the influence of the background is to compare the network performance on several different backgrounds. In Table 5, we compare the MetFluxNet results in three locations: North America, Middle East and North Africa. We obtain very similar results for the three locations, in terms of both RMSE and MAPE. We note that the RMSE and MAPE are slightly higher for North America than for the other two areas. This might be due to the more desertic background we can have in the Middle East and North Africa which usually are less noisy. Moreover, the heterogeneous backgrounds we can find in North America make the estimation more difficult (Roger et al., 2024). The increase in RMSE between North America and the other locations is about 35 kg h$^{-1}$ which represents only a 2.5% increase compared to the results in Middle East and North Africa.

## 4.5 Tests on real data

To validate predictions of our networks, we want to test it on images of real plumes. However, without ground truth, which is generally not available, it is difficult to measure the quality of our prediction. Therefore, we will work with methane plumes observed after the controlled methane releases carried out by Sherwin et al. (2023b) and Sherwin et al. (2023a). In Sherwin et al. (2023b) and Sherwin et al. (2023a), researchers conducted single-blind controlled methane release experiments to evaluate the performance of satellite-based methane detection and quantification methods. They released methane plumes in Arizona between October and November 2021 and October and November 2022. These releases occurred during overpasses of several satellites with methane detection capabilities, including PRISMA and EnMAP. In 2021, three methane plumes were released during PRISMA overpasses. In 2022, one methane plume was released during EnMAP overpasses and three plumes were released during PRISMA overpasses. The plume released on the 27/10/2022, during a PRISMA overpass, is not visible in our methane enhancement retrieval image, the order of magnitude of retrieved concentrations is lower than the background noise level. Hence, we will test our networks on the six remaining plumes for which we have a ground truth.

In Figure 8, we can observe the four of the plumes detected by PRISMA or EnMAP. The plumes have been rotated to be aligned with the x-axis to comply with the alignment pre-processing required for the different versions of MetFluxNet. The red bounding boxes are used for the sparse versions of the networks, and the non-sparse networks used the whole image. In image (d), no pixels needed to be removed; therefore, the bounding box includes the whole image.

In Table 6, we compare the predictions made by MetFluxNet, ProbMetFluxNet and their sparse versions with state-of-the-art methods. Those predictions are provided with $95\%$ confidence intervals. The confidence interval is empirically computed for MetFluxNet and MetFluxNet-sparse. For ProbMetFluxNet and ProbMetFluxNet-sparse, it is computed with the standard deviation estimated by the network. We reproduced the results of S2MetNet (Radman et al., 2023) by training a version of the network on MicroL, the corresponding confidence interval is computed empirically. The work of Sherwin et al. (2023b) and Sherwin et al. (2023a) does not introduce any new methods but gathers the results of different research teams. Therefore, for each plume, we selected the best result obtained among all different teams. To select the best result for a given plume, we considered all the proposed $95\%$ confidence intervals that contain the true source rate and selected the one for which the prediction is closest to the true flux rate. For the four plumes considered here, the best results have been produced with the Integrated Mass Enhancement method (Varon et al., 2018). The $95\%$ confidence intervals were calculated by Sherwin et al. (2023b) and Sherwin et al. (2023a) and the corresponding data and code are publicly available.

For the MetFluxNet network, the ground truth is within the $95\%$ confidence interval for the six plumes. In particular, MetFluxNet makes the best prediction in three cases out of six with predictions very close to the exact value. This shows that false positives, as the ones we can see in Figure 8(b) and (c), do not prevent a good source rate estimation. A possible explanation is the plume alignment: as the position of the plume is fixed in the image, pixels far from it should have a small weight in the final source rate computation. We show in Figure 7 that variations in the background could lead to significant prediction changes. However, when applying bounding boxes, we modify the value of a high number of pixels whereas the brightest false positives visible in Figure 8 represent only a few dozen pixels.

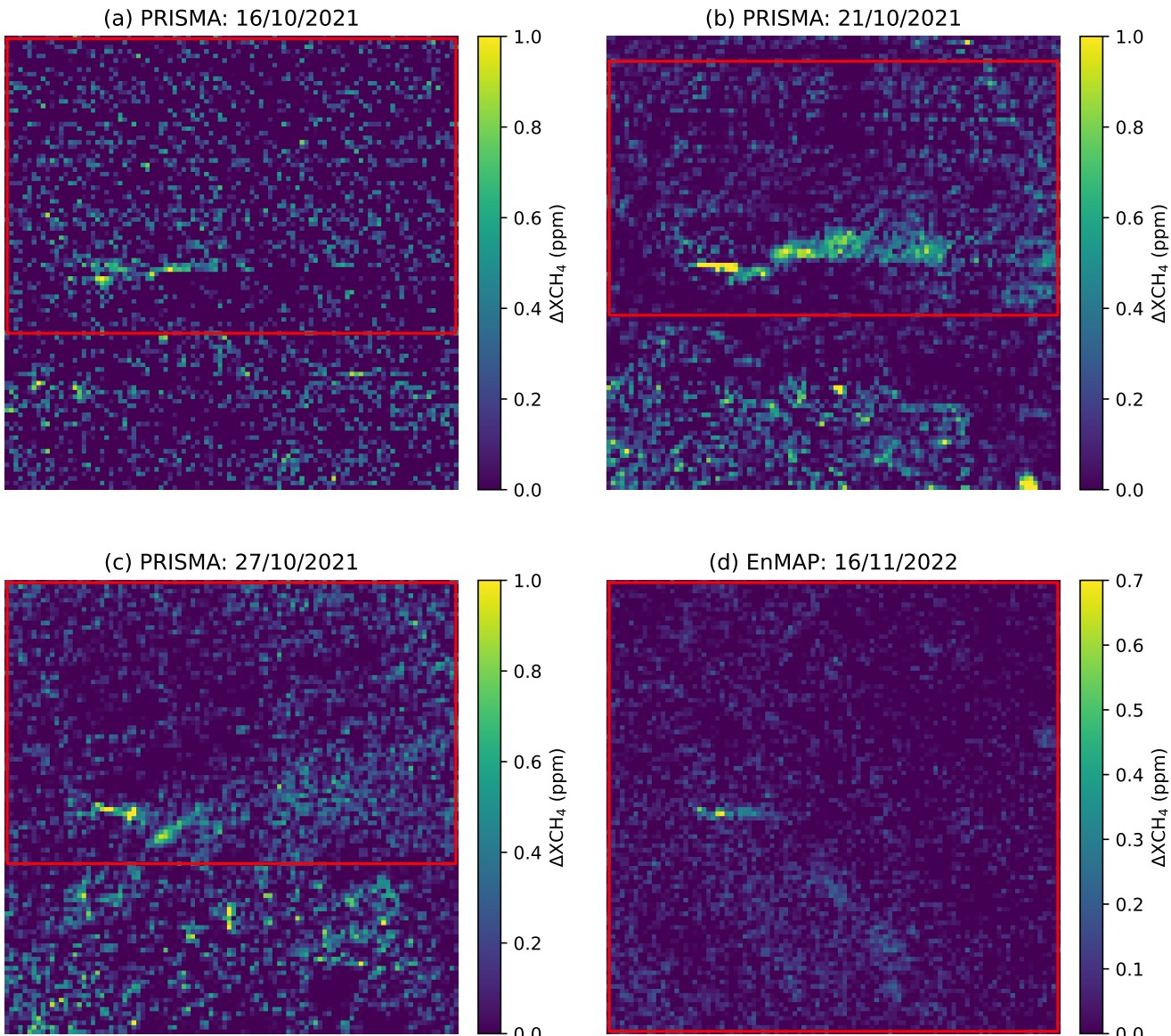

**Figure 8.** Retrieved methane concentration for four methane plumes detected by PRISMA and EnMAP in the methane controlled release experiment of (Sherwin et al., 2023b, a). Pixels outside of the red bounding boxes are removed when using the sparse versions of the networks. The bounding boxes are manually drawn to exclude pixels with high values which do not belong in the plume. The scale is in particles per million (ppm).

It should be noted that even if MetFluxNet can correctly estimate the source rate in the presence of a false positive in the image, it might not be the case if the plume intersects the false positive. Indeed, here the false positive is outside the

**Table 6.** Source rate estimation for plumes detected by PRISMA and EnMAP in the controlled releases experiment of (Sherwin et al., 2023b, a). The source rate values are in kg h$^{-1}$. All the images are from PRISMA, except for the image of the 16/11/2022 which is from EnMAP.

| Network | 16/10/2021 | 21/10/2021 | 27/10/2021 | 07/11/2022 | 16/11/2022 | 30/11/2022 |
|---|---|---|---|---|---|---|
| Best from (Sherwin et al., 2023b, a) (IME) | $3379 \pm 1860$ | $4781 \pm 1854$ | $5051 \pm 2749$ | $\mathbf{388 \pm 686}$ | $1818 \pm 1023$ | $\mathbf{1150 \pm 427}$ |
| S2MetNet (Radman et al., 2023) | $3304 \pm 990$ | $5945 \pm 1416$ | $4666 \pm 1211$ | $1329 \pm 476$ | $1582 \pm 659$ | $1916 \pm 523$ |
| MetFluxNet (ours) | $2735 \pm 798$ | $\mathbf{4695 \pm 985}$ | $\mathbf{3512 \pm 855}$ | $355 \pm 287$ | $\mathbf{1130 \pm 459}$ | $1407 \pm 489$ |
| ProbMetFluxNet (ours) | $2888 \pm 547$ | $4994 \pm 961$ | $4175 \pm 812$ | $633 \pm 166$ | $1255 \pm 270$ | $2355 \pm 486$ |
| MetFluxNet-sparse (ours) | $\mathbf{2569 \pm 728}$ | $5134 \pm 1026$ | $3691 \pm 1048$ | $665 \pm 281$ | $1245 \pm 493$ | $1492 \pm 430$ |
| ProbMetFluxNet-sparse (ours) | $3281 \pm 576$ | $5337 \pm 932$ | $4406 \pm 781$ | $689 \pm 165$ | $1241 \pm 242$ | $2038 \pm 398$ |
| Ground truth | 2355 | 4473 | 3433 | 414 | 1096 | 998 |

plume and does not match the plume alignment criterion. However, if the false positive intersects the plume, it might be interpreted as a local methane concentration maximum which are commonly observed due to eddies. In this case, it can lead to

445 an overestimation of the source rate, as the false positive would increase the apparent total mass of methane.

Overall, the results of MetFluxNet are closer to ground truth than those presented in (Sherwin et al., 2023b, a) for the PRISMA and EnMAP plumes. Even though IME gives the best result for two of the plumes, the ground truth is within the confidence interval produced by MetFluxNet for those cases. Moreover, most of our confidence intervals are also smaller than those of (Sherwin et al., 2023b, a). This shows that MetFluxNet works not only on simulations, but also for real plumes.

**4.6 Comparison with state-of-the-art methods**

To show the improvement brought about by MetFluxNet, we compare it with popular methods to estimate the source rate of point source methane emissions detected with satellite imagery such as IME and S2MetNet (Radman et al., 2023). S2MetNet is a deep learning model based on the EfficientNetV2-L architecture which is then fine-tuned on a simulated dataset generated with LES. Here, we reproduce a version of S2MetNet on MicroL to compare it with MetFluxNet. The methods described here

are tested on the datasets MicroL and S2Test.

The results of the above methods are presented in Table 7. The standard IME version presented here is computed with the effective wind model of Guanter et al. (2021) and the corresponding masking procedure. The IME-MicroL version is computed with our effective wind model and our masking procedure. IME and IME-MicroL give similar results on S2Test but not on MicroL. This is due to the effective wind model of IME-MicroL being fitted on a dataset closer to the test set of MicroL than

460 IME. With the same methodology for IME and an effective model fitted on the train set of MicroL, the standard IME could obtain better results.

We can also observe that both versions of the IME are widely outperformed by deep learning methods. When comparing the deep learning methods, MetFluxNet has a lower RMSE and MAPE than S2MetNet on both datasets. On MicroL, the

**Table 7.** Comparison of different source rate estimation methods. The results are in kg h$^{-1}$ for the RMSE and in percent for the MAPE.

| Method | RMSE (MicroL) | RMSE (S2Test) | MAPE (MicroL) | MAPE (S2Test) |
|---|---|---|---|---|
| IME (Guanter et al., 2021) | 4019 | 3175 | 33.4 | 22.6 |
| IME-MicroL | 1821 | 3339 | 14.2 | 19.5 |
| S2MetNet(Radman et al., 2023) | 1533 | 2280 | 9.7 | 14.0 |
| MetFluxNet (ours) | 1388 | **2255** | **8.3** | **12.7** |
| ProbMetFluxNet (ours) | **1369** | 2377 | **8.3** | 13.4 |

RMSE of MetFluxNet is about 150 kg h$^{-1}$ lower and the MAPE is more than 1% lower. On S2Test, the RMSE of MetFluxNet
and S2MetNet are very close to each other, but in terms of MAPE the gap is the same as on MicroL. This means that Met-
FluxNet significantly outperforms S2MetNet for the low source rates. Moreover, MetFluxNet relies on a much lighter model
than S2MetNet. The ConvNeXtTiny architecture has only 28.6 million parameters whereas EfficientNetV2L has 119 million
parameters. Hence, MetFluxNet is easier to train than S2MetNet and also performs better.

When comparing the results of MetFluxNet on MicroL and S2Test, we see that MetFluxNet performances are worse on
S2Test than on MicroL. The RMSE is about 850 kg h$^{-1}$ higher and the MAPE is 4.4% higher. This can be explained by the
fact that the dataset of Varon et al. (2021) comes from a different simulation setup and is therefore farther from the train set
than the data from our simulations. This difference in RMSE and MAPE does not mean that MetFLuxNet cannot generalize
to different plumes. As we saw in the previous section, it accurately estimated the source rates for the real plumes we tested.
Moreover, our method performs better on S2Test than S2MetNet or the IME, this makes MetFluxNet a method well suited for
real applications.

## 4.7 Robustness to angle variations

The main reason why MetFluxNet outperforms S2MetNet is the plume alignment, which simplifies the source rate estimation
problem. S2MetNet is trained to estimate the source rate of a plume in any direction. This can be useful because it can be
difficult to perfectly align plumes. In Figure 9, we compare the RMSE and MAPE of MetFluxNet and S2MetNet with respect
to the angle between the shape of the plume and the x-axis. We consider angles between $-45°$ and $45°$ as this range represents
a reasonable error range for the plume alignment.

We can observe that MetFluxNet outperforms S2MetNet for all angles tested. In particular, MetFluxNet gives the best results
even when there is a large angle between the plume and the x-axis. This is due to the fact that the training set naturally contains
examples with a non-zero angle between the plume and the x-axis, as explained in Section 3.3.2. Moreover, it is still easier
for MetFluxNet to learn to predict in a range of $140°$ (which is the range of angles in MicroL) than for S2MetNet to learn to

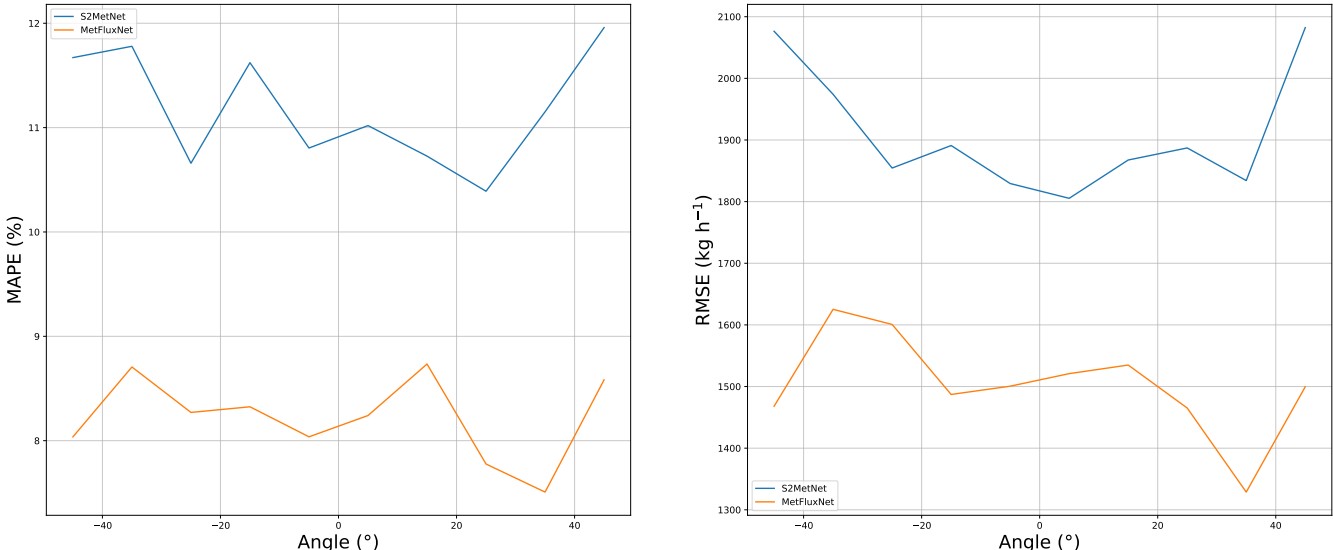

**Figure 9.** Comparison of S2MetNet and MetFluxNet in terms of MAPE and RMSE with respect to the plume angle.

predict for any possible angle. Hence, even without perfect alignment, it is preferable to restrain the plume in a cone rather than allowing unrestricted rotations; the restriction to a cone being easier to satisfy than an exact plume alignment.

We also notice that the MetFluxNet estimation error is rather stable in the range $-45°$ to $45°$. The order of magnitude of the MAPE variation is $0.5\%$. Thus, the prediction quality is the same with an angle of $0°$ and $45°$. There was no overfit of the network around $0°$. Therefore, MetFluxNet is robust to a misalignement of the plumes.

### 4.8 Overfitting when training on MicroS

To show that training with MicroS necessarily leads to overfitting, we compare a network trained on MicroS to a network trained on MicroL. We name MicroSnet and MicroLnet the networks trained on MicroS and MicroL respectively. Both networks are trained without any plume mask, on aligned plumes, and with MSE loss.

In Figure 10, we can compare the results of MicroLnet and MicroSnet on MicroL, MicroS and S2Test. The RMSE and MAPE values are also summarized in Table 8. MicroLnet gives results of the same order of magnitude on MicroL and MicroS. MicroSnet outperforms MicroLnet on MicroS, which was to be expected, but has a higher RMSE and MAPE than MicroLnet on MicroL. In particular, the RMSE of MicroSnet almost triples between MicroS and MicroL.

MicroSnet performs well on MicroS because the train set and the test set are too similar. As we saw in Figure 2, with a 10 s time step, the test set contains plumes that are practically identical to those found in the train set. Even if MicroSnet is trained on more samples than MicroSnet (according to Table 2), it generalizes poorly on MicroL because the training samples are too similar. On the other hand, MicroLnet has RMSE and MAPE of the same order of magnitude on MicroL and MicroS,

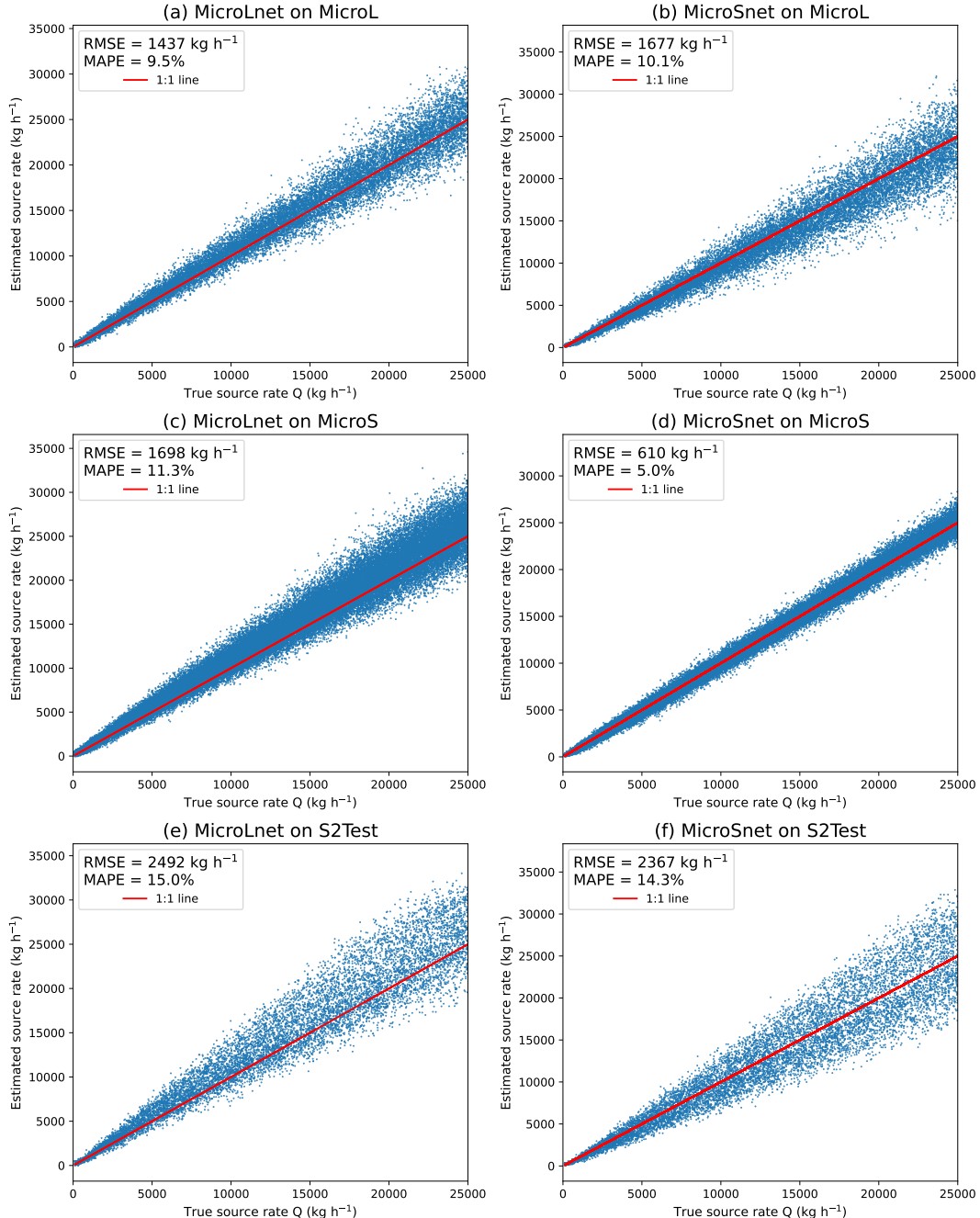

**Figure 10.** Results of the networks MicroLnet and MicroSnet on MicroL, MicroS and S2Test. Each line corresponds to a dataset and each column to a network.

**Table 8.** Comparison between MicroLnet and MicroSnet over three different datasets : MicroL, MicroS and S2Test. The results are in kg h$^{-1}$ for the RMSE and in percent for the MAPE.

| Method | RMSE - MicroL | RMSE - MicroS | RMSE - S2Test | MAPE - MicroL | MAPE - MicroS | MAPE - S2Test |
|--------|---------------|---------------|---------------|---------------|---------------|---------------|
| MicroLnet | **1437** | 1698 | 2492 | **9.5** | 11.3 | 15 |
| MicroSnet | 1677 | **610** | **2367** | 10.1 | **5** | **14.3** |

which shows that the network did not overfit. On S2Test, MicroSnet slightly outperforms MicroLnet but the performance of both networks are way lower than those on their respective test set.

Thus, MicroSnet clearly overfits the MicroS dataset. It performs very well on the test set of MicroS but the performance on this dataset does not correctly represent the ability to quantify source rate under real conditions. Even if the RMSE and MAPE of MicroSnet are of the same order of magnitude as those of MicroLnet when tested on MicroL and S2Test, it is necessary to have a dataset additional to MicroS, to be able to properly evaluate the results of MicroSnet. Therefore, we can simply work with MicroL, as working with MicroS would require using another dataset anyway.

## 5 Conclusions

We introduced MetFluxNet, a new deep learning network for source rate estimation of point source methane emissions detected with the PRISMA and EnMAP satellites. MetFluxNet was trained on MicroL which is a new synthetic plume dataset we generated to train deep learning methods. The use of two different source rate ranges for the train set and the test set of MicroL prevents border effects in the extremes of the testing range. Moreover, the large time gaps chosen for the temporal sampling of the simulated plumes prevents overfit during training.

MetFluxNet can detect a wide range of emissions from 500 kg h$^{-1}$ to 25000 kg h$^{-1}$ and without any wind information or plume labeling. It is based on a ConvNeXtTiny architecture and on an alignment of the plume as pre-processing. We showed that this pre-processing improves the quality of the estimation, in particular in the case of low source rates. The plume alignment also helps to obtain good results even with small network architectures. We showed that MetFluxNet outperforms larger architectures such as EfficientNetV2L thanks to the plume alignment. MetFluxNet achieved a 8.3% in MAPE on our simulated dataset MicroL. It outperforms preexisting methods such as the IME or S2MetNet. We also validated MetFluxNet predictions on real plumes observed in the context of controlled methane release experiments. MetFLuxNet successfully provided 95% confidence intervals for the real plumes we tested.

We also tested variations of the MetFluxNet. We tested ProbMetFluxNet which was designed to provide accurate standard deviation estimations for our predictions. It allowed us to validate the empirical standard deviation estimates computed with the results of MetFluxNet. We also created MetFluxNet-sparse, the purpose of this network was to estimate the source rate after manual false positives removal. MetFluxNet-sparse obtained performances similar to MetFluxNet which shows that a manual

intervention is not needed when working with MetFluxNet. The method presented here was designed for methane plumes, but we aim at generalizing it for other gas or aerosol plumes.

MetFluxNet can be applied to images from other instruments as long as the spatial resolution of the input image is 30 m. This is because, at other spatial resolutions, the visible spatial features of the plumes are not the same. To apply the network to EMIT images, for example, one possible solution would be to apply a super-resolution algorithm to bring the resolution of the EMIT image to 30 m. Another option would be to train a network with images at a spatial resolution of 60 m with the same methodology as described here. However, such a procedure would not guarantee the same results because the lower the spatial

resolution, the fewer spatial features are available for making the source rate estimation.

*Data availability.* EnMAP data are available through the EnMAP planning portal at https://planning.enmap.org/. PRISMA data are available at http://prisma.asi.it/missionselect/. The MicroL dataset is publicly available at https://doi.org/10.5281/zenodo.15618044.

*Author contributions.* EO generated the simulated datasets, designed the method, performed the experiments and wrote the manuscript. TE contributed to setting up the simulations, designing the method and reviewing the manuscript. GF and RM supervised the project, helped

design the method and reviewed the manuscript. EM and JM helped design the method and reviewed the manuscript.

*Competing interests.* The authors declare that they have no conflict of interest.

*Acknowledgements.* The authors thank the CEA for funding this research. The authors also thank Daniel Varon for sharing its simulation data. This work was performed using HPC resources from GENCI–IDRIS (grant AD011012453R3).

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
