# Peer review of "Tightening-up methane plume source rate estimation in EnMAP and PRISMA images"

_EGUsphere, 2025_

## Author Comment (AC1)

**Reply to RC1**

The authors wish to thank the reviewer for their thorough review of the manuscript and their very helpful comments. The comments were carefully considered in the revised version of the manuscript. Below we respond to the different comments and suggestions.

- L18 - The waste and coal mining sectors are also important contributors of the anthropogenic emissions, which can also be controlled/reduced. In the waste case, it generally related to area source emissions. However, the coal mining sector is also related to point source emissions (Karacan, 2025), which is the main focus of this study.

  The mention of the coal sector was added in the text as well as the suggested reference.

- L41 - Please, you could also include here the following reference: Joyce (2023)

  The reference was added.

- L47 - The input are methane concentration maps or methane concentration enhancement maps? Please, clarify.

  The inputs are methane concentration enhancement maps. This was clarified in the text.

- L69-70 - Applying simulations of this kind to L1 data can lead to biases? In other words, can we trust the accuracy of the simulations? I am mostly concerned about how the simulations are convolved to the instrument spectral response function when integrating them into the L1 data. Not applying a correction as in Gorroño (2023) - Eq. 4 might lead to biases in reference to a real-like plume.

  There is indeed a small bias that occurs because we multiply two spectra already convolved with the instrument spectral response function. However this bias depends on the FWHM at the considered wavelength, the smaller the FWHM is, the smaller the bias is. In the case of EnMAP the FWHM is actually quite small with an average value around 12nm in the SWIR. We performed simulations to measure the magnitude of the bias for different methane concentration enhancements. We obtained an average relative error of 0.02% between the spectra convolved correctly and the spectra resulting from the product of two already convolved spectra. This relative error is lower than the SNR of EnMAP in the SWIR. Hence, we consider that the bias is negligible in the case of EnMAP.

- L71 - Later on in the text, it is mentioned that North America locations are more heterogeneous in comparison to the other two. Even with that comment, the location characteristics are vaguely defined. The brightness and heterogeneity of a scene is essential to assess the capability to detect and quantify. It would be very helpful to add further information about it. For instance, a Table listing the sites would be fine. IF the current list of sites are relatively bright and homogeneous, it is important to mention that the performance of the methodology could be worse in darker and more heterogeneous scenes. If so, testing the methods in this kind of scenes would be the ideal way to address this comment. However, mentioning that the location sampling does not consider this kind of scenes and that the results might be worse is also valid.

  We added in the text a list of the different sites as well as an analysis of some of the background samples. In this analysis we show that we have both heterogeneous and homogeneous backgrounds to generate our dataset.

- Section 2 (in general) - A diagram showing the methodology steps would be very useful for the reader. Please, consider to add one. One example of how to do it can be found in Gorroño (2023) - Figure 4.

    A diagram was added to the manuscript.

- L102-107 - Most part of this paragraph is redundant, since the previous paragraph already made this point. Please, consider to remove the redundant parts.

    We reformulated those parts of the text.

- Figure 1,5,6 - Please, add labels in the colorbar (i.e. $\Delta$XCH4)

    Labels in the colorbar were added to all the figures.

- L163 - IME acronym was already defined and Frankenberg paper was already cited. Please, just use IME.

    This was changed in the text.

- L190 - It would be nice to show the Ueff fit. Since the number of points and the wind speed range is larger than in Guanter (2021) simulation dataset, there should be a higher robustness in the fit. Please, consider to add it. On the other hand, it is not explained how the Ueff is deduced. Please, a brief description of the process will be appreciated.

    The authors carefully considered this suggestion, however, we choose not to include the Ueff fit in the manuscript. We believe that it would suggest that there is a perfect or ideal Ueff to reach. However, the Ueff that is fitted depends on the masking procedure and the retrieval algorithm. A different masking procedure for example would lead to a different Ueff to fit and a different robustness of the fit.

    We added in the text that the effective wind model is obtained by linear regression and we described our masking procedure.

- L197 - Frankenberg paper was already cited in the text due to the IME method. Moreover, the IME acronym was already defined. Using the IME acronym would be more consistent than the 'Integrated Mass Enhancement'.

    This was corrected.

- L202 - Please, add a reference to the ConvNext models.

    The reference was added.

- L205 - The validation set should be independent from the training set. Please, clarify.

    The validation set is indeed independent from the training set. This was clarified in the manuscript.

- L211 - Why the shifts have not been done with higher pixel separation (more than 3 pixels)? Please, clarify.

    We used 3 pixels because it seemed empirically a good range for the source location error. In practice, the order of magnitude for the apparent plume source is 2x2 pixels. Hence, it is likely to have errors of this magnitude.

- L213-226: this part is hard to understand. Why is there uncertainty when applying plume rotation? Then, the rotation is only applied for the plumes to follow a x-axis direction? Then, the shift is only 0-3 pixels? Please, I recommend the authors to reword this part of the text to enhance clarity.

The authors thank the reviewer for this remark as this part of the text was not very clear. By "uncertainty" we meant that it is harder for the network to find and extract information from the plume pixels because any pixel can be a plume pixel. When pre-processing an image for inference the image needs to be rotated so that the plume is aligned with the x-axis and then the image needs to be cropped into a 100x100 image with the source in a fixed position. The 0-3 shifts are used in the data augmentation to be robust to imprecision in the image cropping. All of this was clarified in the text.

- L223 - Saying that the plume tail in the retrieval is very noise is not accurate. I would rather say that the authors meant that the plume tail enhancement is approximately at background level. Please, clarify.

This was clarified in the text.

- Section 4.3 - What can be said about the level of uncertainty of the estimations? Since wind speed and plume masking (big error sources) were removed from the calculations, how does it benefit the flux rate uncertainty? Later on the text, when analyzing the controlled releases, it is mentioned that the precision is better. This is an important point because it improves the state of the art situation in which the uncertainty is huge. Please, emphasize this achievement in the text.

A discussion on the level of uncertainties was added in the text.

- L332 - But the main problem was the appearance of retrieval artifacts. What happens if using MetFluxNet on retrievals with false positives? Wouldn't it be better to use MetFluxNet-sparse after cropping the area with the plume to not account for retrieval artifacts? Later on the text, it is shown that the artifacts in the controlled release retrievals do not have a big impact in the results. What would happen if (for instance) there is a facility with a high score in the retrieval? How would the prediction change? This is important because this is a relatively common case in real plumes. Please, discuss.

We added a discussion on this issue in the text.

- L350 - In Sherwin 2023b, there are more PRISMA plumes. Since there are very few plumes from controlled releases to test the methodologies, it is important to leverage the whole extension of available data. Please, consider to extend the analysis to the rest of PRISMA plumes from Sherwin 2023b.

We added to Section 4.5 the other PRISMA plumes, except for the plume released on the 27/10/2022, during a PRISMA overpass. This plume is not visible in our methane enhancement retrieval image, the order of magnitude of retrieved concentrations is lower than the background noise level. This was explained in the text.

- L384 - The IME method is based on the Ueff calibration. This calibration is made using a specific criteria for masking. Here, the masking process is not specified and the lack of accuracy in the IME method could be due to using a masking criteria different from the one used in the original calibration. Please, clarify.

This was clarified in the manuscript. We added a description of our masking procedure. In particular, we ensured that the masking procedure used when reproducing the results of (Guanter et al.,2021) is the same procedure used in their paper.

- Conclusions - It would be great to talk about the implications of this work to other instruments such as EMIT or MethaneSAT. Can these insights be applied to other instruments?

We added a discussion on how and when MetFluxNet could be generalized to other instruments. The main limitation is the spatial resolution which restrains the type and size of spatial features available for the network. Hence, an image from an other instrument needs to be at a 30 m resolution to guarantee a good result when using MetFluxNet.

---

## Author Comment (AC2)

**Reply to RC2**

The authors wish to thank the reviewer for their thorough review of the manuscript and their very helpful comments. The comments were carefully considered in the revised version of the manuscript. Below we respond to the different comments and suggestions.

- PRISMA and EnMAP L1 products are radiance products, and yet L69-70 states that "we inserted those plumes in true EnMAP L1 images to reproduce plumes with real background noise." To my understanding, LES plumes are produced as column enhancement images and added to the true methane retrievals, not to the L1 radiance products. Please correct this description and describe how this combination is implemented.

  We added the description of the procedure used to add the simulation result in the L1 image.

- Introducing realistic background noise is a critical part of the dataset construction, and it is extensively discussed in prior works (Jongaramrungruang et al. 2019, Radman et al. 2023, etc.) However, Section 2.1 only states that 48 background samples were selected from "different locations in North America, Middle East and North of Africa." This is a very broad description for such an important aspect of this work.

  This was clarified in the text with a more extensive presentation of the background with the inclusion of coordinates and a figure containing examples of background samples.

- How many unique 100x100 or 130x130 tiles does "48 background samples" represent, and were they stratified between the training and test datasets? If thousands of unique plumes were inserted into only 48 tiles, I would not be confident that this is "recreat[ing] real conditions as much as possible" (L73-74).

  We have indeed 48 tiles. However, the matched filter standardizes the data when computing the methane enhancement map. Hence, it is sufficient to have variability in the distribution of Delta-XCH4. We can achieve this goal without having all the possible type of backgrounds because different background types can lead to the same matched filter output. We added a figure to illustrate that we have diversity in the distributions of Delta-XCH4. The background diversity used in our work is comparable to other examples in the literature such as (Radman et al., 2023). Nonetheless, the authors agree that saying that it recreates "real conditions as much as possible" can be misleading. We removed the mention of "as much as possible" in the text.

- Why were only EnMAP background images chosen, when the method was evaluated on three PRISMA controlled release plumes? Due to the differences in instruments and spectral resolutions, I would expect background images to look meaningfully different between the two instruments.

  We used EnMAP images because EnMAP is a newer satellite than PRISMA. In particular, the PRISMA satellite is already at the end of its life cycle. To ensure that the model trained here is as sustainable as possible over time, we therefore chose to train only on EnMAP images. However, EnMAP and PRISMA have many similar characteristics, in particular, they have the same spatial resolution. If we compare the enhancement maps calculated from the images of the two satellites, we obtain fairly comparable results. The main difference is that the results obtained with PRISMA are slightly noisier. Metfluxnet is robust to noise as it is trained with different levels of background noise. It is therefore able to make predictions on PRISMA images.

- Please be more precise in which locations the background samples were taken from. Within North America alone, the background can be significantly different between agricultural sites in California, oil and gas infrastructure in the Permian Basin, and coal mines in Pennsylvania.

  This was clarified in the text in the new version of the presentation of the background samples.

- Any further statistical characterization or visualization of the retrievals in the background samples would be very helpful to complete this characterization. For example, if the distribution of the noise and false enhancements present in these tiles are always separable than that of the distribution of plume enhancements, then the posed problem may be too easy compared to those present in real observations (e.g. Figure 6b).

  We added a figure to characterize the background samples as a part of the response to the previous comments. For the issue of false enhancements, MetFluxNet is robust to false enhancements thanks to the plume alignment. A false enhancement far from the expected location of the plume has a low impact on the predicted source rate.

- The discussion of temporal sampling in LES datasets is a critical one and I am glad this was included. While sampling with longer time gaps does reduce their correlation, I disagree that they can be considered "independent," which suggests no correlation at all. This is especially the case for lower wind speeds, and it is later described as such in L121-122. It would also be rigorous to avoid introducing samples from the same simulation to both the training and test sets (see minor comment for L85-86). If this is not possible, authors should justify why.

  The authors agree that talking of independence can mislead the reader and this was removed from the text.

  The dataset was randomly split from the combined 14640 plumes and not from the 244 simulations. This ensures that for each wind speed, we have plumes simulated with different temperature profiles in the train set. We added this explanation in the text.

  The authors are aware that such a process creates correlation between the train set and the test set. However, the S2Test dataset is available here, which provides an independent dataset.

- In Section 3.2, the coefficient of the effective wind model fit to the MicroL dataset in Eq. 6 is significantly different from the one reported in Guanter et al. 2021 (i.e. half). Guanter et al. states that "This relationship was derived using large-eddy simulations specifically performed for a spatial resolution and $\Delta XCH4$ retrieval precision compatible with the PRISMA data." Does this suggest that the simulations performed for the MicroL dataset has significant differences with expected PRISMA observations? Is the same effective wind model also used to provide the baseline emission rate estimate for the EnMAP-observed controlled release plume, and is this appropriate? Since effective wind speed is a critical component for IME, I believe the discrepancy between IME and IME-MicroL deserves further explanation.

  The difference between the two effective wind models stems from the fact that the methods used to calculate the plume mask are different. As explained in (Guanter et al., 2021), a change in the calculation of the mask requires a change in the effective wind model. This point has been clarified in the text. We have also added the method used to obtain our plume mask. The effective wind model described in (Guanter et al., 2021) is the one that provides the IME results in Sherwin et al,2023. For the controlled releases we considered that the ground truth is the baseline and not the estimation from Sherwin et al, 2023.

- In Section 3.3, authors state that their use of the neural network "removes the variability associated with the manual labeling of the plume." However, the authors have introduced a major source of variability in their methodology that requires manual labeling, without much analysis of their sensitivity to the resulting emission estimation. One of the major contributions of this work is aligning all plumes with the x-axis as described in Section 3.3.2. On L213, the authors state that randomly rotating the plume (as is commonly done in prior work) "artificially increases the difficulty of the task," and that randomly shifting the source location (as is common) "adds uncertainty to the position of the plume." The authors follow on to argue that "the uncertainty in the position of

the plume is likely to affect the quality of the source rate estimate," and instead choose to align all plumes with the x-axis and to keep the source location identical. They justify this by stating that "This alignment step can be performed automatically or manually, as most methane plume detection methods rely on the intervention of a human annotator." This is a confusing decision, as it was stated only two paragraphs prior that their method removes the variability associated with manual labeling. Relying on this level of alignment, where the plume source must be at the same exact pixel, and the plume direction exactly aligned, critically couples their model to the quality of the manual labeling for any observed plume. It is true that, despite active research in the field, most missions rely on manual annotation for plume detection. This does not mean, however, that plume source identification, or plume directional alignment, is reliable or consistent. Plume source identification often relies on referencing high-resolution optical basemaps for infrastructure identification, which is only accurate when the source comes from an obvious source visible from the surface (e.g. a controlled release site), and not, for example, a pipe buried underground. Plume direction alignment is even more challenging, since, as the authors describe themselves at the end of Section 3.2, wind data is often low resolution or inaccurate. Observed plumes often display changes in wind speed and direction across both time and altitude, making it unlikely that one "correct" directional alignment even exists. Therefore, this type of plume source and direction alignment is only possible with Large Eddy Simulations, where both information is inherently known. The intent of randomizing these two factors during training is to help the model generalize to real-world observations where both information is unknown (and hence, more difficult). Even in some LES simulations (such as the "Initial image" and 4 m/s frame in Figure 1), the alignment is unclear, and the plume appears diagonal. The analysis of this in Section 4.1 is further misleading. The "+align" model will perform better on its own test dataset than the "+rotation" model will on its test dataset because the task itself is easier. Table 2 and Figure 2 are misleading because they compare models' RMSE and MAPE on different test datasets with either rotated/shifted plumes or aligned plumes, depending on the model. The two models should be evaluated on both test datasets for a direct comparison (as in Table 3); I hypothesize that the "+rotation" model will perform better on the "+align" test dataset than the inverse, demonstrating better generalizability. There are two possible resolutions to this issue. First, the authors could instead present and report on the CNext+rotation model and exchange a minor performance reduction for improved generalizability and robustness for real world scenarios. However, this may require a significant amount of effort to replicate downstream tasks. Second, the authors could additionally present a sensitivity study, correlating the aligned model's performance reduction on datasets with offsets to the plume source position (+- 5 pixels) or rotation (+- 45 deg). This would thoroughly characterize the align model's robustness or brittleness to such noise.

The authors wish to thank the reviewer for this very relevant comment as the plume alignment is a critical part of our method.

There was a lack of clarity on this point in Section 3.3. In practice, the method does not expect the user to know the exact source location and the exact wind angle to align the plume. It is enough to find the apparent source location and plume angle. The apparent source location is usually the vertex of the cone that represents the plume shape. The plume angle is the visible angle of the plume which might differ from the actual wind angle because of eddies.

Nonetheless, when rotating and cropping the plume image, there are uncertainties associated with both operations. Hence, the pre-processing will not always be perfect. That is why we added uncertainties about the location of the source of the plume in the train set. We apply random shifts of the source (from 0 to 3 pixels) in any direction. This ensures that the network will be robust to an inaccurate cropping of the image.

The alignment with the x-axis can also be inaccurate. Therefore, the train set must contain examples of plumes with a non-zero angle with the x-axis. Due to the eddies, the rotations naturally produce during the LES procedure; plumes do not always spread in the same direction as the wind. Hence, the simulated plumes can have a non-zero angle with the x-axis. We can observe this phenomenon in the new Figure 2. For the plumes in the last two columns, the shape of the plume is not aligned with the x-axis. Our train set contains plumes with angles between -70 deg and 70 deg with respect to the x-axis. This ensures that the network will be robust to an inaccurate rotation of the image. Therefore, we do not artificially rotate the simulated plumes to achieve this.

Thus, for the dataset with aligned plumes, we have uncertainties on both source location and plume

alignment.

As suggested, we added in the Experiment Section a discussion on the angle sensitivity. It shows that MetFluxNet has stable performance between -45deg and 45deg and outperforms S2MetNet (trained with all possible rotations) in this range of angles.

We also clarified this in Section 3.3.

On the evaluation of the "+align" model and "+rotation" model the goal was to show that the problem is in fact easier and that is why we compared each model on its corresponding dataset. This fact seems intuitive but it does not necessarily lead to a gain in performance. We show this when we discuss the case of the high source rates, we observe that at high source rates the "+align" and "+rotation" models have the same performance.

- In Section 4.4, authors state that "When estimating the source rate, we first want to remove those false positives before giving the image to the network" (L308). As described in Section 3.2, the IME method accomplishes this with a manually-annotated plume mask. The authors instead implement a different manual background mask that removes the top and bottom edges of the image containing background noise. During training, they "draw random bounding boxes that include the entire plume and remove the pixels outside of it" (L322). Once again, this methodology is only possible because the plumes are generated with LES, and it is known a priori exactly which pixels contain a plume. In an observed plume, the method would have to rely on an accurate, manual plume mask to determine how much of the background edges to remove. In common cases where plume bodies or tails intersect false enhancements caused by noise or surface confusers, this would remain a subjective process that can potentially affect the source rate estimation. Nevertheless, I believe this is an interesting experiment, and it is reassuring that MetFluxNet outperforms MetFluxNet-sparse for three of the four controlled release experiments. In my opinion, this methodology is given too much emphasis, and gives the impression that such manual intervention is necessary for SOTA performance, especially in Figure 6 with the conspicuous red boxes. I suggest that the authors consider splitting Section 4.4 between a new Section 3.3.3 (since background masking is a dataset preprocessing step) and a results section after 4.5, such that the primary presented method requires minimal manual intervention (other than the alignments previously discussed). However, this is subjective, and it could remain as is with stronger emphasis on describing this as an interesting tangent.

  There was also a lack of clarity in the part of the text.

  The random bounding boxes are used during training to be robust to different bounding box sizes and therefore different levels of sparsity in the background distribution. The idea behind the use of bounding boxes is that it is easier to draw a bounding box than a fine mask of the plume.

  We clarified this point in the text. We also added at the end of Section 4.4 the fact that MetFluxNet-sparse is not the main focus paper and we emphasized the drawbacks of the method. In particular, MetFluxNet-sparse is not a part of the final table comparing our methods with the state-of-the-art.

  We chose not to include the background masking in Section 3 as it would suggest that this pre-processing belong in the final method or that it is one of the main contribution.

- Please consider making the MicroL dataset more widely available (e.g. on Zenodo) than making it available by request. This would be a great contribution for the field.

  Thanks for the recommendation, we publicly released the MicroL dataset on zenodo as suggested.

- For all plots and figures, please confirm the units. "True flux rate Q" should instead have physical units, and colorbars of figures showing plumes should have ppm-m.

  Units have been added to all plots and figures.

- L93-94) What method was used to resample the plumes from 25 m to 30 m?

  We used a cubic spline interpolation. This was clarified in the text.

- L85-86) Splitting the dataset before data augmentation is indeed important – was this performed randomly from the combined 14640 plumes, or stratified from the 244 different simulations? This may be related to comment 1.3.

  The dataset was randomly split from the combined 14640 plumes and not from the 244 simulations. This ensures that for each wind speed, we have plumes simulated with different temperature profiles in the train set. We added this explanation in the text.

  The authors are aware that such a process creates correlation between the train set and the test set. However, the S2Test dataset is available here, which provides an independent dataset.

- L96-97) Can plumes be scaled linearly as described without impacting the sensitivity of the simulations?

  Yes, it is a common process to apply a linear scaling (Varon et al, 2018; Guanter et al, 2021). We also verified the validity of the linear scaling by simulating at different source rates.

- Figure 1) Should concentration units be ppm-m?

  The concentration unit is ppm, as in (Guanter et al., 2021)

- Figure 1) Is the "Initial image" after the warm-up period?

  Yes, all the images we showed are after the warm-up period.

- L155) Please include a short discussion of inter-instrument differences in applying MAG1C to PRISMA and EnMAP. Are there different target absorption spectra between instruments?

  We added the discussion on the differences that occur when applying MAG1C to PRISMA and EnMAP. We have different target spectra as the computation of the target spectrum requires to use the instrument spectral response function.

- L202) Please cite Liu et al. 2022 CVPR for ConvNeXt. It would also be useful to provide the number of learned parameters for ConvNeXt-Tiny, EfficientNetv2-B0, and S2MetNet to compare model complexities.

  The reference was added, as well as the number of parameters for the different architectures.

- L204) Model weights pre-trained on ImageNet are designed for three input channels to capture color information. How was this converted for one input channel (i.e. methane column concentration)?

  We converted the enhancement map into a three channel image by copying the concentration map in each additional channel. This was clarified in the text

- L251-252) I am confused why "at high source rates, the methane concentration looks like the ground truth image." By the "ground truth image" do you mean the LES plumes without any background injected?

  Yes, by "ground truth image" we refer to the output of the LES. This was clarified in the text. By "looks like the ground truth image" we mean that the SNR of a given plume is very high, the noise is therefore negligible with respect to the plume, as in the ground truth.

- L256) It is stated in L233 that plumes with source rates below 500 kg/h will be used for visualization purposes, but not to calculate the MAPE and RMSE. Is this the case in Table 2 or not? Please also clearly demarcate this line in Figure 2, since it is presented alongside Table 2.

  In Table 2 we use source rates above 500 kg/h. This was clarified in the figure description, we also changed the figure to include the 500 kg/h demarcation.

- L295-296) I am not sure what "With CNext+NLL we are not only comparing the predictions, but also the standard deviations" means in this context. How is the RMSE and MAPE being calculated for the CNext+NLL model exactly? It doesn't seem to make much sense to calculate the RMSE on the standard deviations.

  The RMSE and the MAPE are not computed on the standard deviations but only on the source rate prediction. This was clarified in the text. The sentence "With CNext+NLL we are not only comparing the predictions, but also the standard deviations" was removed as it was misleading.

- Figure 5) The removed background pixels are difficult to see, even in high-quality color print. Please use a different colormap (diverging, perhaps?) or clearly highlight the removed region as in Figure 6.

  The figure was changed to highlight the removed area.

- L334) The analysis regarding backgrounds from different areas would be more impactful with more precise descriptions of where the backgrounds were sampled, as previously mentioned.

  A more extensive presentation of the background was added in Section 2 with the inclusion of the precise background locations and a figure containing examples of background samples.

- Table 5) Please also indicate which controlled release experiments were observed by which instrument.

  We added the indication of which plume was observed by which instrument in the Table caption.

- Table 5) Please specify IME for "Best from Sherwin et al."

  This was added in the Table.

- L354) Why were no pixels needed to be removed in image (d)? Some background noise is present, even more so than Figure 5.

  In Figure 5, the goal was to show that the prediction changes when we remove parts of the background. However, we agree that in the chosen image there was no need to remove parts of the background as the noise level was very low compared to the plume.

  We did not remove pixels in image (d) to present an example for MetFluxNet-sparse where the background is unchanged.

- L365) Please specify the method by which the 95% confidence interval was determined from Sherwin et al. How is uncertainty calculated for IME?

  The confidence intervals were already calculated by Sherwin et al. We took the values from their publicly available github. This was clarified in the text. Their uncertainty computation involves uncertainty priors on the wind and simulations to estimate the errors on the other components of the IME.

- L370) I agree with the hypothesis that the plume alignment is allowing the model to learn its own mask. However, what happens if a plume bends downward from its initial flow direction? Will it adversely affect source estimation, or is the information near the plume source sufficient? I'm not sure if there is a way to experimentally answer this question.

  The network is robust to variations in the plume angle. This was clarified in the text as part of the response of major comment 3).

- Table 6) I'm not sure it is fair to include the Guanter et al. IME considering the significantly different effective wind models. It may incorrectly imply an issue with the method itself, instead of with the wind model.

There was an issue with our computation of (Guanter et al.,2021) IME results: they were computed with our masking procedure instead of Guanter et al.'s masking procedure. This was corrected and the results have been modified accordingly. The new results corresponding to the IME of Guanter et al. show lower RMSE and MAPE. We chose to keep the IME results of (Guanter et al.,2021) as they are similar to those of IME-MicroL on S2Test, showing that the method is valid. We clarified in the text that the results of IME (Guanter et al.,2021) on MicroL are due the effective wind model and that another effective wind model fitted on the train set of MicroL would give better results.

- L376) Please use a word other than "precision" – accuracy, better metrics, lower error, etc.

  This was corrected in the text.

- L389) Please include these parameter numbers earlier in L202.

  The parameters numbers were added when the different networks are first cited.

- L400, Figure 7) These are interesting results, but I believe the performance metrics would be better digested with an additional table to accompany the figures (or describe the actual numbers in the text)

  A Table summarizing the results was added.